# Geometrically Constrained Stenosis Editing in Coronary Angiography via Entropic Optimal Transport

**Jialin Li** [1]  **Zhuo Zhang** [1]  **Yue Cao** [1]  **Guipeng Lan** [1]  **Jiabao Wen** [1]  **Xiao Shuai** [1]  **Jiachen Yang** [1]

## Abstract

The scarcity of high-quality imaging data for coronary angiography (CAG) stenosis limits the clinical translation of automated stenosis detection. Synthetic stenosis data provides a practical avenue to augment training sets, improving data quality, diversity, and distributional coverage, and enhancing detection precision and generalization. However, diffusion-based editing commonly relies on soft guidance in a noise-initialized reverse process, offering limited pixel-level precision and structure preservation. We propose the **OT-Bridge Editor**, which reframes localized editing as a constrained entropic optimal transport (OT) problem and leverages geometric information to steer the generation path, enabling stronger geometric control. Extensive experiments show that our synthesized angiograms consistently improve downstream stenosis detection, yielding substantial relative gains of 27.8% on the public ARCADE benchmark and 23.0% on our multi-center dataset, supported by consistent qualitative results.

## 1. Introduction

CAG varies substantially across centers and scanners, and precise stenosis annotation is costly in expert effort and time, resulting in persistently limited high quality training and evaluation data. Realistic synthesis of stenosis lesions offers a practical route to expand training sets and improve detector generalization. However, to be useful for downstream measurement and diagnosis, the synthesis process must support local, geometrically precise edits while preserving vessel topology and continuity. Diffusion based image editing methods casts image editing as conditional

[1]School of Electrical and Information Engineering, Tianjin University, Tianjin, China. Correspondence to: Zhuo Zhang <z_zhuo@tju.edu.cn>, Shuai Xiao <xs611@tju.edu.cn>.

*Proceedings of the 43rd International Conference on Machine Learning*, Seoul, South Korea. PMLR 306, 2026. Copyright 2026 by the author(s).

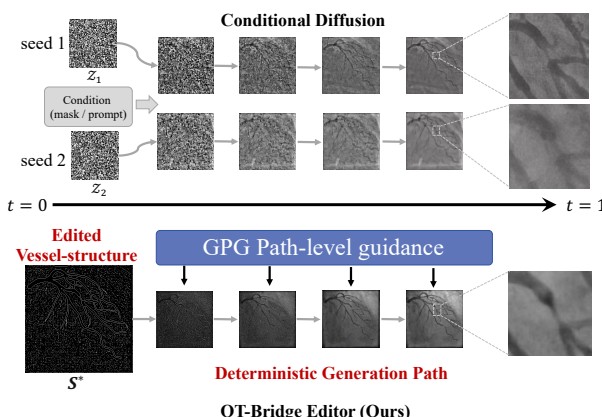

*Figure 1.* Conceptual comparison between conditional diffusion and OT-Bridge Editor. OT-Bridge Editor fixes the start state $S^*$ (edited vessel-structure) and enforces hard, path-level guidance (GPG) within a Schrödinger-bridge formulation, producing a deterministic generation path toward the target.

generation, where an input image is provided together with editing conditions (Zhang et al., 2023) such as textual instructions (Brooks et al., 2023), binary masks (Lugmayr et al., 2022), semantic segmentations (Park et al., 2019), or explicit geometric constraints (Mou et al., 2024), and the model samples an output from a target conditional distribution. While highly successful at producing perceptually plausible and semantically controllable edits, this paradigm offers limited guarantees on spatial accuracy (Hertz et al., 2022), including boundary alignment. First, conditioning is typically enforced through guidance terms or auxiliary inputs, which act as soft constraints during denoising and may be insufficient to precisely lock geometry. Second, most editors sample from a noise initialized reverse process even though the input image already contains reliable anatomy outside the intended edit region. This makes it difficult to strictly preserve non-target content and to localize changes to the desired area. In essence, the desired operation is not to reconstruct the whole image from noise, but to transport (Su et al., 2022; Kim et al., 2023; Zhang et al., 2025b) the source image toward a target that differs only locally.

A direct approach is to initiate the generative process from the given source image and construct a diffusion bridge that connects the source and the desired edited output (De Bortoli et al., 2021; Wang et al., 2024). This viewpoint is closely aligned with OT, since stenosis editing can be interpreted as transferring probability mass from the source to a target edited distribution while minimizing unnecessary change (Léonard, 2013). However, classical OT does not directly specify a pathwise generative mechanism (Peyré & Cuturi, 2019). Moreover, in high-dimensional spaces, solving OT under strong geometric constraints can be numerically fragile (Peyré & Cuturi, 2019; Vacher et al., 2021; Lan et al., 2025), limiting its direct use for controllable medical image synthesis.

In this work, we reformulate geometrically constrained image editing as a constrained entropic OT problem and propose a solution based on Schrödinger bridges (SB). Motivated by prior contour-domain editing work (Frangi et al., 1998; Elder & Goldberg, 1998), we perform editing in a designed Vessel-structure Composite Domain. Specifically, we transformed the geometric editing specifications into the boundary conditions of the corresponding Schrödinger system, and achieved deterministic generation path control through geometric generation-path guidance (GPG). This design enables pixel-accurate boundary control in the edited region while explicitly preserving structural integrity in unedited regions, making the synthesized CAG suitable as augmentation data for training stenosis detectors.

Such fine grained control over lesion shape and placement increases morphological and positional diversity, improving coverage of clinically relevant variations. We demonstrate pixel accurate edits with strong clinical realism, and show that detectors trained with our synthesized angiograms achieve substantial performance gains on the public AR-CADE benchmark (Popov et al., 2024), with consistent gains observed on our multi center dataset.

**Our Contributions** We summarize the contributions. **Algorithm** We propose OT-Bridge Editor, transformed the image editing problem with geometric constraints into a constrained entropy OT problem and solved it using the SB; furthermore, we propose geometric generation-path guidance, imposed explicit geometric condition constraints on the generation path to achieve supervision at the generation path level, thereby achieving pixel-level boundary alignment within the editing area. **Application** We instantiated the CAG stenosis editing framework to generate synthetic vascular images with clinical authenticity and pixel accuracy. Using synthetic data, we trained multiple stenosis detectors on public benchmarks and multi-center clinical datasets, achieving significant improvements.

## 2. Related Work

**Synthetic Data for Medical Vision Tasks** Recent work on medical synthetic data and augmentation primarily addresses limited high-quality annotations, privacy and sharing barriers and class imbalance (Koetzier et al., 2024; Khosravi et al., 2025). One line studies cross-modal synthesis (Caetano et al., 2025; Koetzier et al., 2024) and evaluates how generative models support downstream tasks across modalities such as MRI/CT/PET (Wang et al., 2025; He et al., 2025; Dayarathna et al., 2024; Cao et al., 2026; Pan et al., 2025). Another line directly synthesizes image-label pairs with diffusion or foundation models, improving segmentation (Dorjsembe et al., 2024; Qiu et al., 2025b), classification (Zhang et al., 2025a), and detection (Oh & Jeong, 2023; Nazir et al., 2025). Work on controllable synthesis explores lesion inpainting with spatial masks (Konz et al., 2024), tunable stenosis augmentation for CAG (Seo et al., 2025), and coronary morphology generation for digital twins (Guo et al., 2025). However, existing methods still struggle to deliver pixel-accurate control of lesion location and morphology (Guo et al., 2025).

**Image Editing with Structural Constraints** Early conditional image-to-image transformation works used paired supervision to directly map the structural input to the image, such as pix2pix (Isola et al., 2017), pix2pixHD (Wang et al., 2018) and SPADE (Park et al., 2019). Diffusion-based editors instead inject conditions into the denoising process. For semantic structure guidance, representative methods such as SDEdit (Meng et al., 2021), SDM (Wang et al., 2022), and more recent controller-based frameworks like ControlNet (Zhang et al., 2023) and Uni-ControlNet (Zhao et al., 2023) incorporate mask-based generation during the diffusion process. SiameseDiff based on ControlNet (Qiu et al., 2025a) improve medical downstream applicability. Nevertheless, a recurring limitation of existing structure-controlled editing methods is that they primarily emphasize semantic consistency rather than precise geometric morphology control (Zhang et al., 2023).

**Optimal Transport and Schrödinger Bridge for Generative Modeling** OT offers a geometric framework for matching distributions by finding cost-minimizing couplings between marginals (Kantorovich, 2006; Villani, 2021). In practice, entropic regularization yields tractable solvers with Sinkhorn iterations and improved numerical stability (Cuturi, 2013; Peyré & Cuturi, 2019; Lan et al., 2026). Schrödinger Bridge (SB) extends OT to stochastic processes by seeking a path distribution that matches prescribed endpoint marginals while remaining close to a reference diffusion (Schrödinger, 1932; Léonard, 2013). This perspective has directly enabled OT/SB-based generative modeling. Diffusion Schrödinger Bridge (DSB) provides a diffusion-based approximation to iterative proportional fit-

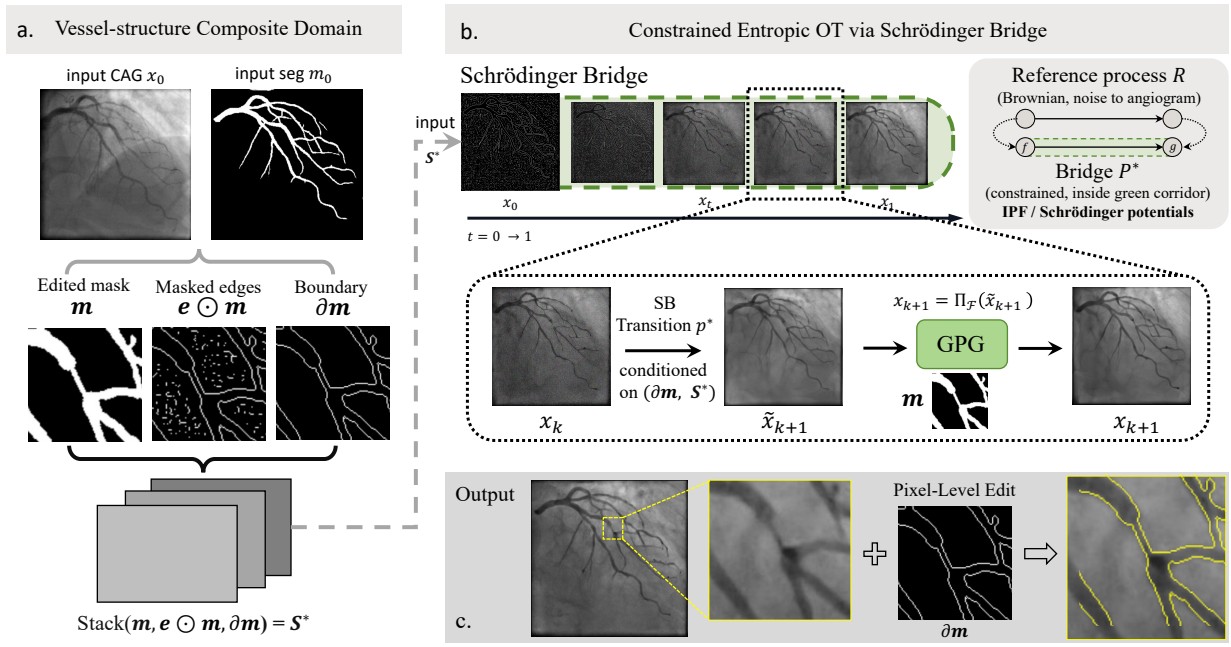

*Figure 2.* Overview of OT-Bridge Editor. (a) We build a vessel-structure composite start state $S^*$ from an edited mask $m$ and its geometric cues (edges and boundary). (b) A constrained diffusion Schrödinger bridge implements entropic OT under hard geometric constraints, producing a bridge process $P^*$ relative to the reference process $R$; each step combines an SB transition with path-level guidance (GPG) and a projection $\Pi_{\mathcal{F}}$ to enforce feasibility. (c) The output angiogram realizes the specified stenosis and achieve the pixel-level edit.

ting, scaling SB to high-dimensional data (De Bortoli et al., 2021). DSB-style models have been validated for unconditional generation and distribution learning (De Bortoli et al., 2021), as well as conditional generation and image-to-image restoration/translation, where bridging from a source distribution often improves structure preservation (Liu et al., 2023; Li et al., 2023). Beyond natural images, SB/DSB formulations have been applied to structured medical vision tasks such as ambiguous segmentation (Baru et al., 2025) and to constrained transitions in engineering domains, e.g., global routing with objective-oriented gradient feedback (Shi et al.). Meanwhile, many SB/DSB methods still underexploit aligned or paired data, and better leveraging alignment to improve convergence and controllability remains an active direction (Somnath et al., 2023).

## 3. OT-Bridge Editor

### 3.1. Preliminaries and OT-Bridge Editor Overview

We study geometry-constrained stenosis editing for CAG. Given an input angiogram $\mathbf{x}_0 \in \mathbb{R}^{H \times W}$ and an editing specification, our goal is to generate an edited angiogram $\mathbf{x}_1$ whose geometry follows the specification within an editing region, while the vascular identity and non-target appearance remain unchanged outside this region.

We define a binary editing mask $\mathbf{M} \in \{0,1\}^{H \times W}$ and its

complement $\bar{\mathbf{M}} = \mathbf{1} - \mathbf{M}$. The specification further induces a target geometric description $\mathbf{S}^\star$ via a structural operator $\mathcal{S}(\cdot)$, such as a lumen mask or boundary representation. These definitions separate narrow, geometry-specific edits in $\mathbf{M}$ from strict preservation in $\bar{\mathbf{M}}$, which is critical for thin and connected vascular structures.

Figure 2 outlines the OT-Bridge Editor pipeline. Starting from $\mathbf{x}_0$, we first construct $(\mathbf{M}, \mathbf{S}^\star)$ from the editing specification, then cast stenosis editing as constrained entropic OT between a source distribution anchored at $\mathbf{x}_0$ and a target distribution restricted by a geometric feasibility set. We solve this constrained transport using a Schrödinger Bridge(SB), obtaining a bridge process and an entropic interpolation trajectory, and finally apply GPG to enforce explicit geometric conditions along the generation path.

### 3.2. Constrained Entropic OT Editing via Schrödinger Bridge

#### 3.2.1. FROM GEOMETRIC EDITING TO CONSTRAINED ENTROPIC OT

Let $\mathbf{s}$ denote the transported state, where $\mathbf{s} = \mathbf{x}$ in pixel space and $\mathbf{s} = \mathbf{z}$ in latent space. We model stenosis editing as transporting probability mass from a source distribution $\mu_0$ anchored at the input to a target distribution $\mu_1$ restricted by geometric feasibility. In our main setting, $\mu_0 = \delta_{\mathbf{s}_0}$ with

**Algorithm 1** OT-Bridge Editor (overview)

1: **Input:** angiogram $\mathbf{x}_0$; editing specification (location, severity, shape); steps $T$.
2: **Output:** edited angiogram $\mathbf{x}_1$.
3: Construct editing mask $\mathbf{M}$ and protected region $\bar{\mathbf{M}}$; derive target geometry $\mathbf{S}^\star$.
4: Set editing space: $\mathbf{s} = \mathbf{x}$.
5: Formulate geometry-constrained editing as constrained entropic OT between $(\mu_0, \mu_1)$ with feasibility set $\mathcal{F}$.
6: Solve the OT via a Schrödinger Bridge to obtain a bridge process $\{\mathbf{s}_k\}_{k=0}^T$.
7: **for** $k = 0$ to $T - 1$ **do**
8:  Propagate one bridge step to obtain $\tilde{\mathbf{s}}_{k+1}$.
9:  Apply GPG to enforce geometric conditions along the path: $\mathbf{s}_{k+1} \leftarrow \mathrm{GPG}(\tilde{\mathbf{s}}_{k+1}; \mathbf{M}, \mathbf{S}^\star, \mathbf{x}_0)$.
10: **end for**
11: Decode if needed: $\mathbf{x}_1 = \mathbf{s}_T$.

$\mathbf{s}_0 = \mathbf{x}_0$.

Given an editing mask $\mathbf{M}$ and its complement $\bar{\mathbf{M}}$, we define a feasibility set that enforces strict preservation outside the editing region and geometric compliance inside it,

$$\mathcal{F} = \left\{ \mathbf{x} \;\middle|\; \mathbf{x} \odot \bar{\mathbf{M}} = \mathbf{x}_0 \odot \bar{\mathbf{M}}, \; \mathcal{S}(\mathbf{x} \odot \mathbf{M}) = \mathbf{S}^\star \right\}, \quad (1)$$

where $\mathcal{S}(\cdot)$ extracts the target geometry (e.g., lumen mask or boundary), and $\mathbf{S}^\star$ is induced by the editing specification. We consider entropic OT with additional constraints,

$$\min_{\pi \in \Pi(\mu_0, \mu_1)} \langle \mathbf{C}, \pi \rangle + \varepsilon \, \mathrm{KL}(\pi \| \mathbf{K}) \quad \text{s.t.} \quad \pi \in \mathcal{Q}, \quad (2)$$

where $\mathbf{C}$ is induced by a mask-aware cost, $\mathbf{K}$ is a reference coupling kernel, and $\mathcal{Q}$ collects hard geometric restrictions used by our bridge construction.

### 3.2.2. COST DESIGN FOR NARROW EDITS AND STRUCTURE PRESERVATION

To reflect narrow, geometry-specific edits, we adopt a mask-aware cost that discourages changes in the protected region,

$$c(\mathbf{x}, \mathbf{y}) = \left\| (\mathbf{x} - \mathbf{y}) \odot \bar{\mathbf{M}} \right\|_2^2 + \lambda_M \left\| (\mathbf{x} - \mathbf{y}) \odot \mathbf{M} \right\|_2^2. \quad (3)$$

In our framework, strict preservation outside $\mathbf{M}$ is enforced by feasibility (Eq. 1), while the cost regularizes the magnitude of edits and stabilizes transport inside $\mathbf{M}$.

### 3.2.3. SCHRÖDINGER BRIDGE AS A DYNAMIC SOLVER

Directly solving Eq. 2 in high-dimensional spaces is challenging. We therefore compute the constrained entropic transport via a Schrödinger Bridge (SB), which lifts the problem to path space. Let $R$ be a reference Markov process on trajectories $\{\mathbf{s}_t\}_{t \in [0,1]}$ (implemented as a discretized diffusion chain). SB computes the closest path measure to $R$ in KL divergence while matching endpoints,

$$P^\star = \arg \min_{P: P_0 = \mu_0, \, P_1 = \mu_1} \mathrm{KL}(P \| R), \quad (4)$$

yielding an entropic interpolation trajectory and bridge dynamics that expose the full generation path.

### 3.2.4. PRACTICAL BRIDGE SAMPLING PROCEDURE

Starting from $\mathbf{s}_0$, we roll out the bridge process for $T$ steps to obtain $\mathbf{s}_T$,

$$\tilde{\mathbf{s}}_{k+1} \sim p^\star(\mathbf{s}_{k+1} \mid \mathbf{s}_k), \qquad k = 0, \ldots, T - 1. \quad (5)$$

We decode the terminal state if needed,

$$\mathbf{x}_1 = \begin{cases} \mathbf{s}_T, & \mathbf{s} = \mathbf{x}, \\ \mathcal{D}(\mathbf{s}_T), & \mathbf{s} = \mathbf{z}. \end{cases} \quad (6)$$

This section establishes the constrained OT formulation and SB-based solver that produce a transport path from $\mu_0$ to $\mu_1$. The next subsection introduces Geometric Generation-Path Guidance (GPG) to impose explicit geometric supervision along the bridge trajectory.

### 3.3. Geometric Generation-Path Guidance with Path-Level Geometric Supervision

#### 3.3.1. DEFINITION OF GPG

We introduce Geometric Generation-Path Guidance (GPG) as a path-level supervision mechanism applied to the bridge trajectory (Algorithm. 2). Let $\{\mathbf{x}_k\}_{k=0}^T$ denote the generated states in pixel space. GPG acts on intermediate states and enforces geometric feasibility during rollout, rather than only at the endpoint.

In this work, we instantiate GPG as a step-wise projection onto a geometric feasibility set. Given $\mathcal{F}$ (Eq. 1), we define a projection-like operator $\Pi_{\mathcal{F}}(\cdot)$ and update each bridge step by

$$\tilde{\mathbf{x}}_{k+1} \sim p^\star(\mathbf{x}_{k+1} \mid \mathbf{x}_k), \qquad k = 0, \ldots, T - 1, \quad (7)$$

$$\mathbf{x}_{k+1} \leftarrow \Pi_{\mathcal{F}}(\tilde{\mathbf{x}}_{k+1}). \quad (8)$$

The operator $\Pi_{\mathcal{F}}$ is designed to promote two complementary goals: geometric compliance inside the editing region and stability outside the region.

**Geometric projection with protected-region consistency.** We implement $\Pi_{\mathcal{F}}$ by solving a constrained (or penalized) projection problem that matches the target stenosis geometry inside the mask while discouraging changes in the protected region:

$$\Pi_{\mathcal{F}}(\mathbf{x}) = \arg \min_{\mathbf{y}} \; \mathcal{L}_{\text{geo}}(\mathbf{y}) + \lambda_{\text{out}} \mathcal{L}_{\text{out}}(\mathbf{y}),$$

$$\mathcal{L}_{\text{geo}}(\mathbf{y}) = \mathcal{D}_{\text{geo}}\big(\mathcal{S}(\mathbf{y} \odot \mathbf{M}), \mathbf{S}^\star\big), \quad (9)$$

$$\mathcal{L}_{\text{out}}(\mathbf{y}) = \left\| (\mathbf{y} - \mathbf{x}_0) \odot \bar{\mathbf{M}} \right\|_2^2.$$

GPG constrains the entire generation path

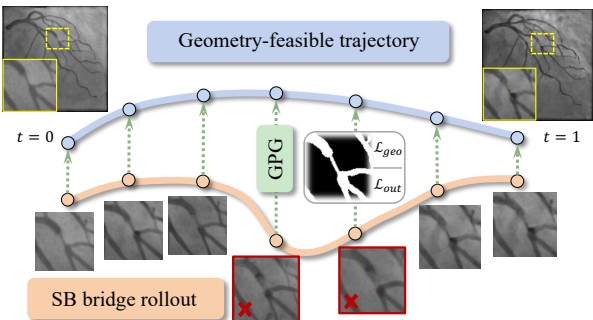

**Algorithm 2** OT-Bridge Editor with Geometric Generation-Path Guidance (GPG)

1: **Input:** angiogram $\mathbf{x}_0$; mask $\mathbf{M}$; target geometry $\mathbf{S}^\star$; steps $T$.
2: **Output:** edited angiogram $\mathbf{x}_1$.
3: Initialize state $\mathbf{s}_0 = \mathbf{x}_0$.
4: **for** $k = 0$ to $T - 1$ **do**
5:      Sample one SB bridge step: $\tilde{\mathbf{s}}_{k+1} \sim p^\star(\mathbf{s}_{k+1} \mid \mathbf{s}_k)$.
6:      Decode if needed: $\tilde{\mathbf{x}}_{k+1} = \tilde{\mathbf{s}}_{k+1}$.
7:      Apply GPG: $\mathbf{x}_{k+1} \leftarrow \Pi_{\mathcal{F}}(\tilde{\mathbf{x}}_{k+1})$ using Eqs. 9.
8:      Re-encode if needed: $\mathbf{s}_{k+1} = \mathbf{x}_{k+1}$.
9: **end for**
10: Output $\mathbf{x}_1 = \mathbf{x}_T$.

*Figure 3.* GPG constrains the entire SB rollout. Starting from an unconstrained SB trajectory (orange), intermediate states may drift and introduce off-target changes (red boxes). GPG applies step-wise geometric supervision at each bridge step (green arrows), pulling the path into a geometry-feasible corridor (blue) that preserves non-target anatomy while enforcing the desired stenosis geometry inside the editing region (yellow insets).

where $\mathcal{S}(\cdot)$ extracts the geometric descriptor, $\mathcal{D}_{\text{geo}}(\cdot, \cdot)$ measures geometric discrepancy, and $\lambda_{\text{out}}$ controls protected-region stability. This step-wise correction injects explicit geometric supervision along the bridge path and mitigates trajectory drift on thin vessels. Implementation details are provided in Appendix A.1.

### 3.3.2. SUPERVISION SIGNALS AND GEOMETRIC OPERATORS

GPG requires geometric discrepancy measures that are reproducible and aligned with vascular morphology. We use boundary-based supervision to achieve pixel-level control over stenosis shape and location. Let $\partial\mathbf{S}(\mathbf{x})$ denote the boundary extracted from $\mathcal{S}(\mathbf{x})$. We implement $\mathcal{D}_{\text{geo}}$ using signed distance transforms (SDT) for stability. Let $B_M(\mathbf{x}) = \partial\mathbf{S}(\mathbf{x} \odot \mathbf{M})$ and $\phi^\star = \text{SDT}(\partial\mathbf{S}^\star)$. We define

$$\mathcal{D}_{\text{geo}}\big(B_M(\mathbf{x}), \partial\mathbf{S}^\star\big) = \frac{1}{|B_M(\mathbf{x})|} \sum_{\mathbf{u} \in B_M(\mathbf{x})} \big|\phi^\star(\mathbf{u})\big|. \quad (10)$$

Alternative choices include Chamfer distance between boundary point sets and Boundary IoU computed on thin boundary bands. When centerline or radius constraints are available, they can be incorporated by adding consistency terms on the extracted centerline $\mathcal{C}(\cdot)$ or radius profile $r(\cdot)$ within $\Omega_M$.

## 4. Experiment and Analysis

### 4.1. Datasets and Setups

**Datasets.** We used real CAG data to train our OT-Bridge Editor, and used both real and synthetic CAG data for downstream task validation. The real data includes the publicly

available **ARCADE** dataset and the **multi-center internal dataset** we constructed. The specific details of the datasets are provided in Appendix A.2.1.

**Metrics.** Our metrics include (i) editing quality and (ii) downstream detection performance; detailed definitions and computation protocols are provided in Appendix A.2.4.

**Baselines.** We evaluate whether controllable stenosis synthesis improves downstream detection by fixing the detector backbone and training schedule and varying only the synthesis method used to generate the augmented set. We compare OT-Bridge Editor against representative controllable generators for batch augmentation, including conditional GANs (Pix2PixHD (Wang et al., 2018), SPADE (Park et al., 2019)) and diffusion-based editing/augmentation methods (SDEdit (Meng et al., 2021), SDM (Wang et al., 2022), SiameseDiff (Qiu et al., 2025a), and the task-specific DiGDA (Seo et al., 2025)); full settings and protocol details are provided in Appendix A.2.2.

**Hyperparameter settings.** For OT-Bridge Editor, we run the bridge rollout for $T = 50$ steps and use entropic regularization $\varepsilon = 1 \times 10^{-2}$ to control transport smoothness. We apply GPG at every 5 step ($K = 5$). In the GPG projection, we set the protected-region stability weight to $\lambda_{\text{out}} = 10$ and the geometric alignment term uses the SDT-based boundary discrepancy in Eq. 10 with weight $\lambda_{\text{geo}} = 1$. Complete values are provided in Appendix A.2.3.

### 4.2. Geometric Constraint CAG Stenosis Editing

We use the vascular structure as a geometric constraint, perform CAG stenosis editing in the Vessel-structure Composite Domain and synthesize a corresponding angiogram. We compare our results with representative baselines.

**Controllability of stenosis location and morphology.** We evaluate how well each method follows segmentation-defined target stenosis specifications. We edit masks to shift stenosis along the vessel and vary its shape, and generate one angiogram per target. Fig. 4 visualizes location-controlled

*Table 1.* Quantitative result for editing geometric constraints of CAG Stenosis. We compare vessel mask $\hat{M}$ obtained from edited CAG with target mask $M_{\text{tgt}}$ provided as the geometric constraint.

| Method | mIoU↑ | Dice↑ |
|---|---|---|
| Pix2PixHD (Wang et al., 2018) | $0.801_{\pm0.066}$ | $0.621_{\pm0.051}$ |
| SPADE (Park et al., 2019) | $0.726_{\pm0.071}$ | $0.595_{\pm0.083}$ |
| SDEdit (Meng et al., 2021) | $0.781_{\pm0.062}$ | $0.645_{\pm0.077}$ |
| SDM (Wang et al., 2022) | $0.783_{\pm0.042}$ | $0.644_{\pm0.055}$ |
| SiameseDiff (Qiu et al., 2025a) | $0.837_{\pm0.052}$ | $0.722_{\pm0.068}$ |
| **OT-Bridge Editor(ours)** | $\mathbf{0.892}_{\pm0.037}$ | $\mathbf{0.774}_{\pm0.055}$ |

edits with zoomed-in ROIs, showing pixel-accurate insertion and minimal drift outside the target region; additional qualitative results are provided in Appendix B (Fig. 9). Tab. 1 quantifies geometric compliance by comparing the extracted vessel mask $\hat{M}$ with the target constraint $M_{\text{tgt}}$. OT-Bridge Editor achieves the highest compliance (mIoU $0.892 \pm 0.037$, Dice $0.774 \pm 0.055$), outperforming SiameseDiff and other baselines.

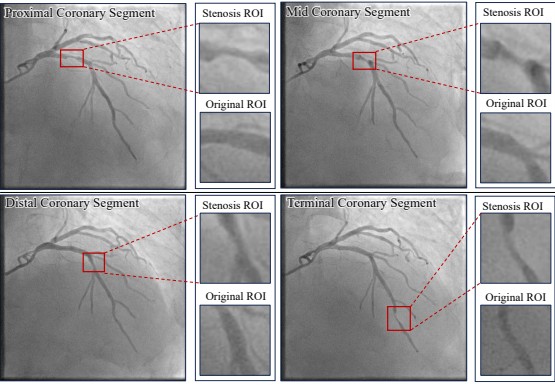

*Figure 4.* Qualitative results for location and shape controlled stenosis editing. We shows an edited CAG for a target stenosis position in the Proximal, Mid, Distal, and Terminal segments. The red box marks the target region, and the right column provides zoomed views of the edited Stenosis ROI and the corresponding Original ROI. The edits producing pixel-accurate stenosis insertion while preserving vessel appearance outside the target region.

**Image quality comparison.** We evaluate edited angiogram quality using representative realism and fidelity metrics in Tab. 2. OT-Bridge Editor achieves the best realism and structure fidelity. Consistently, Fig. 5 shows that our results have cleaner stenosis boundaries and less drift. As can be seen from the figures and tables in the Fig.6 and Tab. 11 in Appendix.B, the same conclusion is also reflected in the downstream tasks.

### 4.3. Detection Performance Evaluation

We evaluate downstream stenosis detection on the same real test splits and compare a diverse set of detectors, in-

*Table 2.* Representative image-quality metrics for segmentation-conditioned CAG editing. We report distribution-level realism (FID↓, IS↑) and paired fidelity to the original image (LPIPS↓, SSIM↑). Full metrics are reported in Appendix B.

| Method | FID↓ | IS↑ | LPIPS↓ | SSIM↑ |
|---|---|---|---|---|
| Pix2PixHD (Wang et al., 2018) | 52.874 | 4.150 | 0.704 | 0.676 |
| SPADE (Park et al., 2019) | 78.636 | 2.831 | 0.600 | 0.577 |
| SDEdit (Meng et al., 2021) | 46.900 | 3.120 | 0.410 | 0.705 |
| SDM (Wang et al., 2022) | 39.417 | 2.708 | 0.485 | 0.616 |
| SiameseDiff (Qiu et al., 2025a) | 34.200 | 2.710 | 0.281 | 0.790 |
| **OT-Bridge Editor (ours)** | **16.747** | **4.63** | **0.248** | **0.878** |

cluding YOLO (Jocher et al., 2023), DINO-DETR (Zhang et al., 2022), Grounding DINO (Liu et al., 2024) and RT-MDet (Lyu et al., 2022). To isolate the effect of synthesized data, for each detector we adopt three training settings: *Real-only* (trained on the real training split of the target benchmark), *Synth-only* (trained only on our synthesized training set with inherited labels), and *Real+Synth* (trained on the union of real and synthesized training samples). All models are evaluated on the same real validation/test splits (no synthetic images are used for evaluation). For stochastic training pipelines, we repeat each experiment three times and report the mean results.

*Table 3.* Synthetic data has enhanced the detection performance in **ARCADE**. Full results are in Appendix B (Tab. 10).

| Detector | Training | mAP@0.5↑ | F1↑ |
|---|---|---|---|
| YOLOv8 | Real-only | $0.525_{\pm0.009}$ | $0.664_{\pm0.009}$ |
| | Synth-only | $\underline{0.662}_{\pm0.008}$ | $\underline{0.737}_{\pm0.008}$ |
| | Real+Synth | $\mathbf{0.727}_{\pm0.006}$ | $\mathbf{0.775}_{\pm0.007}$ |
| DINO-DETR | Real-only | $0.508_{\pm0.010}$ | $0.645_{\pm0.010}$ |
| | Synth-only | $\underline{0.615}_{\pm0.012}$ | $\underline{0.697}_{\pm0.011}$ |
| | Real+Synth | $\mathbf{0.720}_{\pm0.007}$ | $\mathbf{0.766}_{\pm0.008}$ |
| Grounding DINO | Real-only | $0.276_{\pm0.012}$ | $0.453_{\pm0.014}$ |
| | Synth-only | $\underline{0.330}_{\pm0.015}$ | $\underline{0.505}_{\pm0.015}$ |
| | Real+Synth | $\mathbf{0.418}_{\pm0.011}$ | $\mathbf{0.564}_{\pm0.013}$ |
| RTMDet | Real-only | $0.545_{\pm0.008}$ | $0.687_{\pm0.009}$ |
| | Synth-only | $\underline{0.625}_{\pm0.010}$ | $\underline{0.726}_{\pm0.010}$ |
| | Real+Synth | $\mathbf{0.675}_{\pm0.007}$ | $\mathbf{0.749}_{\pm0.008}$ |

**Evaluation in ARCADE and multi-center.** First, we conducted experiments on the ARCADE dataset to measure the gains brought by the synthetic data, and summarized the results in Tab. 3. Then, we repeated the same detector suite and training settings and evaluated on our multi-center internal dataset (Appendix A.2.1) to assess the generalization ability across centers (Tab. 4). Among all the detectors, *Real+Synth* typically provides the best overall performance, while *Synth-only* demonstrates how a model based solely on the synthetic distribution can effectively transfer its performance to real test data.

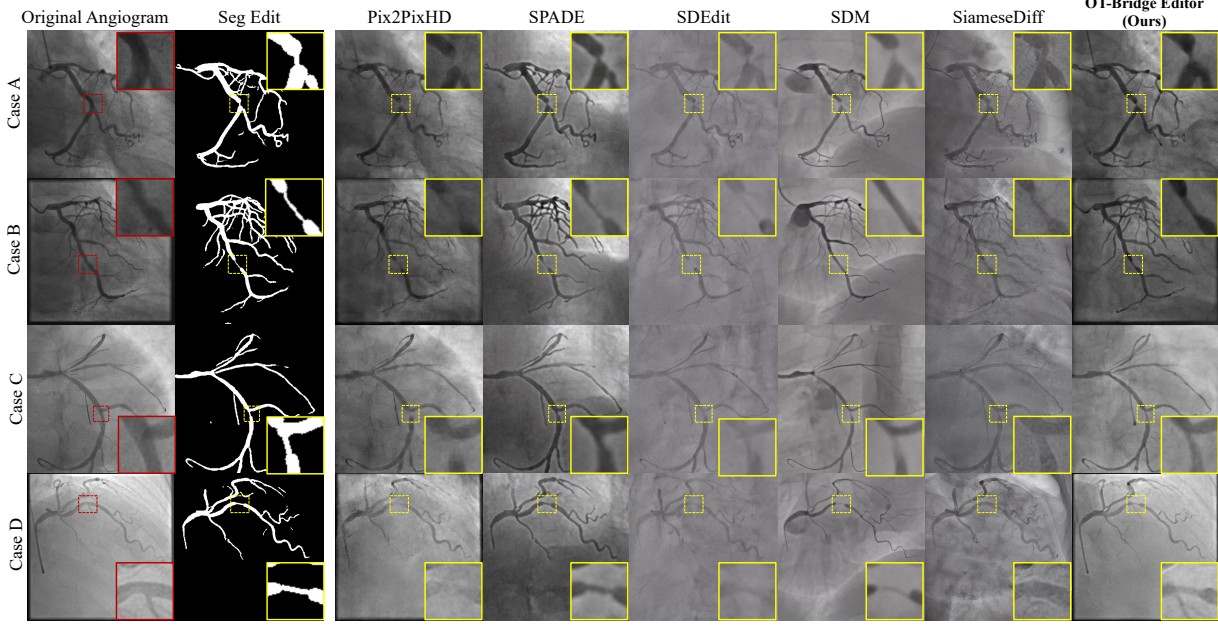

*Figure 5.* Qualitative comparison of geometry-constrained stenosis editing on CAG. For six representative cases (A-D), we show the input CAG and the desired structural change specified by an edited vessel mask, followed by results from baselines and our OT-Bridge Editor. Red boxes indicate the target region on the original image; yellow boxes highlight the same region across methods.

*Table 4.* Synthetic data consistently improves detection performance on the **multi-center dataset**, indicating strong cross-center generalization. Full results are in Appendix B (Tab. 10).

| Detector | Training | mAP@0.5↑ | F1↑ |
|---|---|---|---|
| YOLOv8 | Real-only | $0.654_{\pm0.011}$ | $0.725_{\pm0.010}$ |
| | Synth-only | $0.582_{\pm0.014}$ | $0.648_{\pm0.013}$ |
| | Real+Synth | $\mathbf{0.731}_{\pm0.008}$ | $\mathbf{0.779}_{\pm0.007}$ |
| DINO-DETR | Real-only | $0.638_{\pm0.010}$ | $0.710_{\pm0.009}$ |
| | Synth-only | $0.565_{\pm0.013}$ | $0.635_{\pm0.012}$ |
| | Real+Synth | $\mathbf{0.725}_{\pm0.007}$ | $\mathbf{0.768}_{\pm0.008}$ |
| Grounding DINO | Real-only | $0.385_{\pm0.012}$ | $0.532_{\pm0.014}$ |
| | Synth-only | $0.312_{\pm0.016}$ | $0.485_{\pm0.015}$ |
| | Real+Synth | $\mathbf{0.442}_{\pm0.010}$ | $\mathbf{0.588}_{\pm0.011}$ |
| RTMDet | Real-only | $0.615_{\pm0.009}$ | $0.695_{\pm0.010}$ |
| | Synth-only | $0.548_{\pm0.012}$ | $0.622_{\pm0.011}$ |
| | Real+Synth | $\mathbf{0.688}_{\pm0.006}$ | $\mathbf{0.754}_{\pm0.007}$ |

*Table 5.* Ablation on editing domain and structure preservation. Alignment is measured in the edited region (higher is better). Fidelity is measured outside the mask (higher is better for SSIM; lower is better for LPIPS).

| Setting | Edited region ↑ | | Outside mask | |
|---|---|---|---|---|
| | Dice | mIoU | SSIM↑ | LPIPS↓ |
| Edge | $0.592_{\pm0.107}$ | $0.483_{\pm0.095}$ | $0.533_{\pm0.070}$ | $0.472_{\pm0.027}$ |
| Seg | $0.767_{\pm0.067}$ | $0.632_{\pm0.072}$ | $0.694_{\pm0.016}$ | $0.329_{\pm0.027}$ |
| Comp | $\mathbf{0.892}_{\pm0.037}$ | $\mathbf{0.774}_{\pm0.055}$ | $\mathbf{0.878}_{\pm0.030}$ | $\mathbf{0.248}_{\pm0.007}$ |
| C w/o P | $0.802_{\pm0.061}$ | $0.674_{\pm0.075}$ | $0.786_{\pm0.058}$ | $0.317_{\pm0.006}$ |

## 4.4. Ablation Study

### 4.4.1. WHY COMPOSITE-DOMAIN EDITING ENABLES PRECISE LESION CONTROL WITH STRUCTURE PRESERVATION?

To verify that editing in the vessel-structure composite domain enables both precise lesion control and vessel structure preservation, we ablate the editing domain and the structure-preservation constraint. Specifically, we compare single edge map domain editing, single segmentation domain editing, and composite-domain editing that uses a edge map and a vessel edge map as the generation starting point, and we further remove the non edited region preservation constraint under the composite setting to test its necessity. All variants use the same target mask and stenosis control conditions. We report Boundary Dice/IoU within the edited region to measure boundary alignment, and SSIM/LPIPS outside mask to measure structure fidelity and connectivity in the non-edited region. Results in Tab. 5 show that full composite-domain setting achieves the best boundary alignment while producing the smallest changes outside mask, indicating that composite representation and explicit preservation constraint are both important.

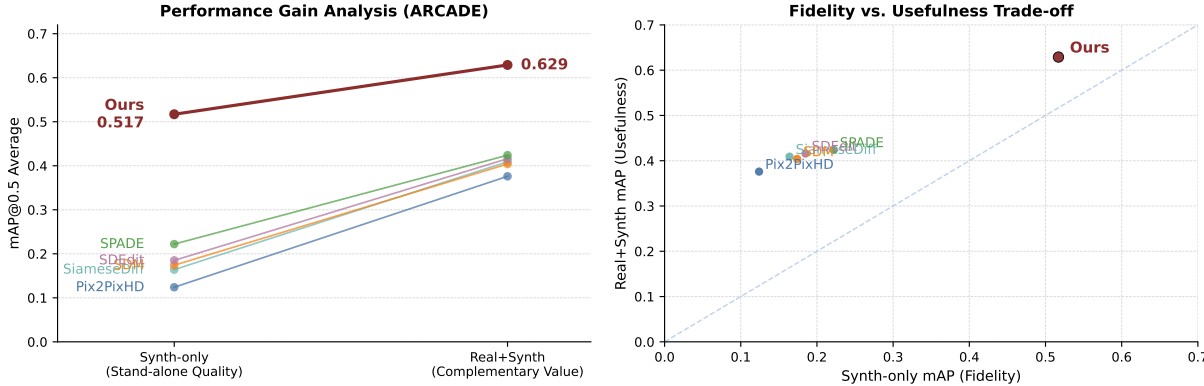

*Figure 6.* Performance gain analysis on ARCADE. We compare *Synth-only* mAP@0.5 (stand-alone fidelity) and *Real+Synth* mAP@0.5 (complementary usefulness), showing that OT-Bridge Editor achieves the best trade-off and the strongest overall gains.

#### 4.4.2. WHY GEOMETRIC GENERATION-PATH GUIDANCE ACHIEVES PIXEL-LEVEL EDITING?

To isolate the effect of geometric generation-path guidance (GPG), we ablate whether geometric constraints are applied along the generation path or only at the endpoint. Specifically, we compare an endpoint constraint variant that enforces the target geometry only at the final step with our full GPG that imposes hard geometric constraints throughout the bridge trajectory. All variants share the same target lesion specification. We quantify pixel-level editing using boundary Dice/IoU metrics, and we plot the trajectory error curve defined as the distance to the target boundary over generation steps. Results in Tab.6 show that full GPG achieves the lowest final boundary error and the smoothest, consistently low trajectory error across steps.

*Table 6.* Ablation of geometric generation-path guidance (GPG). We compare enforcing geometric constraints only at the endpoint versus along the entire bridge trajectory. We report boundary alignment (higher is better) and trajectory error statistics (lower is better), where $\mathcal{E}_t$ measures the distance to the target boundary at step $t$.

| Variant | bDice↑ | bIoU↑ | $\mathcal{E}_T \downarrow$ | $\overline{\mathcal{E}} \downarrow$ |
|---|---|---|---|---|
| Endpoint | $0.765_{\pm 0.024}$ | $0.682_{\pm 0.028}$ | $2.8_{\pm 1.2}$ | $8.5_{\pm 2.1}$ |
| w/o $\partial m$ | $0.582_{\pm 0.015}$ | $0.455_{\pm 0.018}$ | $12.4_{\pm 1.8}$ | $13.1_{\pm 2.0}$ |
| GPG (Ours) | $\mathbf{0.895}_{\pm 0.008}$ | $\mathbf{0.812}_{\pm 0.009}$ | $\mathbf{1.1}_{\pm 0.3}$ | $\mathbf{1.3}_{\pm 0.4}$ |

#### 4.4.3. SCALING SYNTHETIC AUGMENTATION: HOW MUCH SYNTHETIC DATA IS ENOUGH?

To answer how much synthetic data is sufficient for improving stenosis detection, we perform a data-scaling ablation that varies the amount of OT-Bridge Editor synthesized CAG while keeping the real training set fixed. Concretely, let $N_{\text{real}}$ denote the number of real training images. We generate synthetic sets with ratios $r \in 0, 0.1, 0.25, 0.5, 1.0, 2.0$,

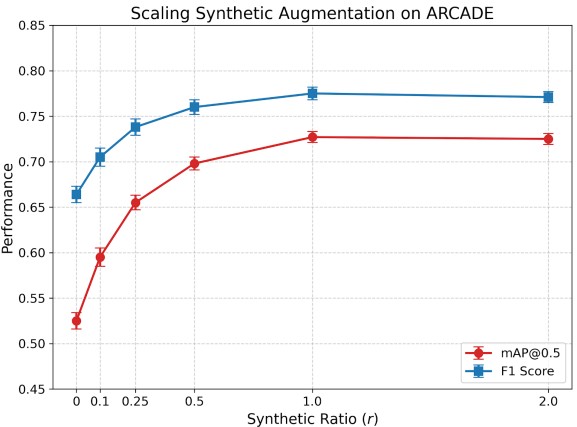

*Figure 7.* Scaling synthetic augmentation on ARCADE. Detection performance (mAP@0.5 and F1) improves as the synthetic ratio $r$ increases, with gains saturating around $r=1.0$ (1:1 Real+Synth).

*Table 7.* Robustness under imperfect vessel-mask supervision. We perturb the editing mask by boundary jitter, erosion/dilation, and spatial displacement, and evaluate both geometric editing quality and downstream detector performance.

| Mask perturbation | Dice↑ | F1↑ | mAP@0.5↑ |
|---|---|---|---|
| None | 0.774 | 0.737 | 0.662 |
| Boundary jitter | 0.767 | 0.717 | 0.645 |
| Erosion/dilation | 0.772 | 0.642 | 0.587 |
| Spatial displacement | 0.764 | 0.561 | 0.521 |

where $N_{\text{synth}} = r \cdot N_{\text{real}}$. We evaluate on the same held-out test sets using mAP@0.5 and F1. According to the results shown in Fig 7, the gain of the synthetic data reaches saturation around $r = 1$, which is related to the upper limit of the diversity of the synthetic data being dependent on the size of the validation set.

### 4.4.4. HOW DOES PERFORMANCE DEGRADE WITH NOISY OR IMPERFECT VESSEL MASKS?

Because our editing process uses vessel masks as geometric supervision, we further study its sensitivity to imperfect masks. We perturb the supervision masks with three common error types: boundary jitter, erosion/dilation, and spatial displacement. Boundary jitter and erosion/dilation mainly simulate local contour errors, whereas spatial displacement simulates localization mismatch between the mask and the target vessel region. We then evaluate the generated results using the same editing-quality metric and the same downstream detector protocol. As shown in Tab. 7, our method degrades gracefully under mask perturbations. Dice remains relatively stable across all perturbations, suggesting that moderate boundary noise does not substantially disrupt geometric editing quality. In contrast, downstream detection performance drops more noticeably, especially under spatial displacement. This indicates that accurate localization of the edited region is more important than minor boundary inaccuracies, and that the proposed framework remains useful under moderate segmentation noise while benefiting from well-aligned masks.

## 5. Further Discussion and Conclusions

### 5.1. Image Editing as Constrained Optimal Transport

Our formulation views stenosis editing as transporting a given source sample toward a target edited distribution while minimizing unnecessary change. This trajectory-constrained, localized editing yields more stable outputs and better preserves non-target anatomy.

### 5.2. Conclusion

We proposed OT-Bridge Editor, a stenosis editing framework that reformulates geometrically constrained image editing as a constrained entropic OT problem. By operating in a vessel-structure composite domain and guiding the generation trajectory with GPG, the method achieves pixel-level control in edited region. Extensive experiments indicate that the resulting synthetic angiograms are clinically realistic and effective for data augmentation, yielding consistent improvements for stenosis detection on both public and multi-center datasets.

### 5.3. Limitations

Generalization across unseen centers, rare lesion morphologies, and different acquisition protocols should be further studied.

## Impact Statement

This paper aims to advance controllable and geometry-constrained generative modeling for medical image synthesis, focusing on augmenting training data for coronary stenosis detection. The expected benefit is improved robustness and generalization of detection models under data scarcity, supporting research and clinical decision support workflows.

As with most synthetic data approaches, careful validation remains necessary before use in safety-critical settings. We therefore evaluate on held-out real test splits and position the method as a data augmentation tool rather than a diagnostic system.

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

# A. Implementation Details

## A.1. Geometric Generation-Path Guidance Details

GPG can be viewed as a projected dynamics that aligns a Schrödinger-bridge (SB) rollout with a geometric feasibility corridor. As illustrated in Fig. 8, an unconstrained SB transition may drift away from the geometry-feasible set and introduce off-target changes. GPG corrects this behavior by inserting an explicit feasibility enforcement step after each SB transition, i.e., sampling $\tilde{\mathbf{x}}_{k+1} \sim p^\star(\mathbf{x}_{k+1} \mid \mathbf{x}_k)$ followed by $\mathbf{x}_{k+1} \leftarrow \Pi_{\mathcal{F}}(\tilde{\mathbf{x}}_{k+1})$ (Eqs. 7–8, Algorithm 2). This converts endpoint-only conditioning into path-level geometric supervision.

**Feasibility corridor and path-level enforcement.** We define the feasibility concept through two complementary requirements: geometric compliance inside the editing region and stability outside the region. In practice, $\Pi_{\mathcal{F}}$ is instantiated by the projection-like optimization in Eq. 9, where $\mathcal{L}_{\text{geo}}$ enforces the target stenosis geometry via a reproducible discrepancy (e.g., SDT-based boundary distance in Eq. 10), and $\mathcal{L}_{\text{out}}$ penalizes deviations from the original anatomy in the protected region. This construction makes the feasible corridor explicit: intermediate states are continuously pulled toward geometry-consistent configurations rather than relying on a single terminal constraint.

## How GPG acts at one intermediate step

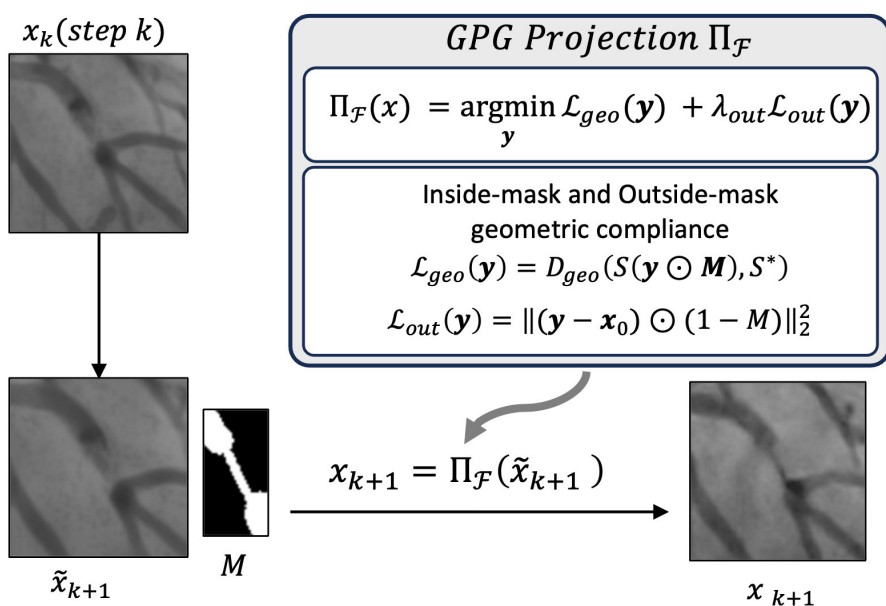

*Figure 8.* How GPG acts at one intermediate step.

## A.2. Experimental protocols

### A.2.1. DATASETS AND HARDWARE FOR EXPERIMENTS

**Real datasets and splits.** We evaluate on two real-world CAG datasets. First, we use the public **ARCADE** benchmark with Train/Val/Test splits of 1,000/100/100 real images. Second, we collect a **multi-center internal dataset** from three difference centers, with Train/Val/Test splits of 5000/500/500 real images. All lesion annotations are stored in the COCO format (Lin et al., 2014). The same real Train split is used to train the OT-Bridge Editor (angiogram–segmentation pairs) and to train the downstream detector, while Val/Test are used only for model selection and final evaluation, respectively.

**Synthesized augmentation set.** To quantify the effect of controllable synthesis on downstream detection, we additionally generate an augmented set of 5,000 edited angiograms using OT-Bridge Editor and use them only for detector training. We

evaluate detectors on held-out real images with Val/Test splits of 200/200. All synthesis is performed by editing real training images with paired segmentation masks, and the edited samples retain the original COCO-style instance annotations with updated lesion geometry after editing.

**Implementation notes and usage.**    For fair comparison across augmentation methods, we keep the real training set fixed and vary only the synthesized augmentation set. When training OT-Bridge Editor, we use angiogram–segmentation pairs derived from the corresponding training splits above. Additional preprocessing, center-wise statistics, and the generation protocol are reported together with other experimental details in Appendix A.2.2.

All models are trained on a single server equipped with four NVIDIA L20 GPUs. Unless otherwise specified, training and inference are performed on these four GPUs with standard mixed-precision acceleration, and all reported results are obtained under the same hardware setting.

### A.2.2. BASELINE DETAILS AND DOWNSTREAM DETECTION PROTOCOL

We follow a controlled evaluation protocol to attribute detection gains to synthesized data. For each synthesis method (ours and baselines), we generate an augmented set and train the same detector under three settings: *Real-only*, *Synth-only*, and *Real+Synth*. All detectors share the same backbone, optimizer, training schedule, and data splits; evaluation is performed on the identical real validation/test splits. Implementation details for each generator (inputs, conditioning, and hyperparameters) are reported in this appendix.

**Common setup for all augmentation generators.**    For every method, we generate paired training samples from the same source images and the same target lesion specifications derived from annotations. Unless a method natively requires a different interface, the inputs consist of the original angiogram and its associated lesion mask (and, when applicable, auxiliary structure maps such as vessel segmentation). All synthesized images are resized to the detector input resolution and saved with consistent intensity normalization. We generate the same number of synthetic samples per training split and keep the sampling seed list fixed across methods.

**Pix2PixHD (Wang et al., 2018).**    *Inputs and conditioning.* Pix2PixHD is trained to map a conditioning tensor to an edited angiogram. The conditioning tensor concatenates (i) a lesion mask, and (ii) a structural map (vessel segmentation). *Architecture.* We use the standard Pix2PixHD generator with a multi-scale discriminator. *Optimization.* We train with Adam and the default Pix2PixHD losses (GAN loss, feature matching, and perceptual loss). *Training schedule.* We train for a fixed number of epochs on the training split with a linear learning-rate decay in the second half. *Inference.* At test time, we apply the trained generator once per conditioning specification to produce the augmented set.

**SPADE (Park et al., 2019).**    *Inputs and conditioning.* SPADE uses spatially-adaptive normalization conditioned on semantic layouts. We provide the lesion mask and vessel segmentation as the semantic layout. *Architecture.* We adopt the standard SPADE generator and multi-scale discriminator. *Optimization and losses.* We use the default SPADE objectives, including adversarial and perceptual components. *Training schedule.* SPADE is trained on the same training split with a fixed schedule matched in total iterations to Pix2PixHD. *Inference.* We synthesize edited angiograms by feeding the target layout and sampling the style code from a fixed seed set.

**SDEdit (Meng et al., 2021).**    *Inputs and conditioning.* SDEdit performs image editing by starting diffusion from a noised version of an input image and applying guidance from a target constraint. We use the real angiogram as the input, and enforce the lesion region using a binary spatial mask. *Diffusion backbone.* We use a pretrained diffusion model (same image domain) and do not finetune it for fairness. *Hyperparameters.* We fix the diffusion step budget and the starting noise level for all images, and tune the mask-guidance strength on the validation split. *Inference.* For each sample, we run the reverse process once to obtain an edited angiogram; all runs use the same step count and seed list.

**SDM (Wang et al., 2022).**    *Inputs and conditioning.* SDM conditions diffusion on a semantic map. We provide the lesion mask as the semantic condition and optionally include vessel segmentation as an additional channel. *Training/finetuning.* We finetune SDM on the training split using paired (conditioning, image) data, with all other settings held fixed across methods. *Hyperparameters.* We use a fixed diffusion step count, classifier-free guidance probability, and guidance scale tuned on validation. *Inference.* We generate one edited image per condition using the same seed list.

*Table 8.* Distribution of synthesized stenosis edits in the synthetic training set ($N$=5,000) across vessel branch, lesion location, and stenosis severity.

| Attribute | Bin | # | % |
|---|---|---:|---:|
| Branch | LAD | 2300 | 46.0 |
| | LCX | 1150 | 23.0 |
| | RCA | 1300 | 26.0 |
| | LM | 150 | 3.0 |
| | Other / side branch | 100 | 2.0 |
| Location | Proximal | 1700 | 34.0 |
| | Mid | 2200 | 44.0 |
| | Distal | 1100 | 22.0 |
| Severity (%DS) | 25–49 (mild) | 1700 | 34.0 |
| | 50–69 (moderate) | 1850 | 37.0 |
| | 70–99 (severe) | 1300 | 26.0 |
| | 100 (occlusion) | 150 | 3.0 |

**SiameseDiff ([Qiu et al., 2025a](#)).** *Inputs and conditioning.* SiameseDiff is an exemplar-based diffusion editor. We use the original angiogram as the source exemplar and construct a target specification by combining the lesion mask with the desired severity/shape parameters when supported. *Training/finetuning.* We follow the official training recipe, finetuning on the training split with paired exemplars and edit targets. *Hyperparameters.* We match the diffusion step count and guidance schedule to the other diffusion baselines, and tune the edit strength on validation. *Inference.* We synthesize the augmented set by running the editor once per sample with fixed seeds and step counts.

**Detector training protocol.** We use the same stenosis detector across all augmentation methods. For each setting (*Real-only*, *Synth-only*, *Real+Synth*), we train the detector from the same initialization and keep the optimizer, learning rate schedule, batch size, and number of epochs fixed. Early stopping and model selection are performed on the real validation split only. All reported detection metrics are computed on the identical real test split.

### A.2.3. DETAILED HYPERPARAMETER SETTINGS

This section details the hyperparameter configurations for our proposed OT-Bridge Editor and the baseline methods used in the comparative experiments. All models were trained and evaluated on a server equipped with four NVIDIA L20 GPUs using mixed-precision acceleration.

**OT-Bridge Editor (Ours)** We solve the constrained entropic optimal transport problem using a Schrödinger Bridge (SB) formulation. The specific hyperparameters for the bridge rollout and the Geometric Generation-Path Guidance (GPG) are as follows:

- **Bridge Rollout Steps ($T$):** We set the total number of time steps for the bridge process to $T = 50$.

- **Entropic Regularization ($\epsilon$):** To control the smoothness of the transport plan, we use an entropic regularization coefficient of $\epsilon = 1 \times 10^{-2}$.

- **GPG Frequency ($K$):** We apply the Geometric Generation-Path Guidance projection at every $K = 5$ steps during the generation process.

- **Protected-Region Stability Weight ($\lambda_{out}$):** In the GPG projection (Eq. 9), the weight governing the preservation of the non-edited region is set to $\lambda_{out} = 10$.

- **Geometric Alignment Weight ($\lambda_{geo}$):** The weight for the geometric alignment term, calculated using the Signed Distance Transform (SDT) boundary discrepancy (Eq. 10), is set to $\lambda_{geo} = 1$.

### A.2.4. METRIC DETAILS

**Notation.** Let $\mathbf{x}$ denote a real image and $\hat{\mathbf{x}}$ a synthesized/edited image. Let $\mathcal{D}_r$ and $\mathcal{D}_s$ be the real and synthesized sets. For segmentation masks, let $A, B \subseteq \Omega$ be foreground pixel sets (or their binary masks) on image domain $\Omega$, and let TP, FP, FN

denote pixel-wise true/false positives/negatives. For detection, a predicted bounding box (or rotated box) is matched to a ground-truth box using IoU thresholding.

**FID.** We report the Fréchet Inception Distance (FID) between synthesized and real distributions by extracting features with an Inception-style network and fitting Gaussians to features from $\mathcal{D}_r$ and $\mathcal{D}_s$. Let $(\boldsymbol{\mu}_r, \boldsymbol{\Sigma}_r)$ and $(\boldsymbol{\mu}_s, \boldsymbol{\Sigma}_s)$ be the empirical mean and covariance of features, then

$$\text{FID}(\mathcal{D}_r, \mathcal{D}_s) = \|\boldsymbol{\mu}_r - \boldsymbol{\mu}_s\|_2^2 + \text{Tr}\left(\boldsymbol{\Sigma}_r + \boldsymbol{\Sigma}_s - 2(\boldsymbol{\Sigma}_r\boldsymbol{\Sigma}_s)^{\frac{1}{2}}\right). \tag{11}$$

Lower is better.

**IS.** We report the Inception Score (IS) computed from the conditional label distribution $p(y|\hat{\mathbf{x}})$ produced by an Inception-style classifier:

$$\text{IS}(\mathcal{D}_s) = \exp\left(\mathbb{E}_{\hat{\mathbf{x}}\sim\mathcal{D}_s}\left[\text{KL}\big(p(y|\hat{\mathbf{x}}) \,\|\, p(y)\big)\right]\right), \quad p(y) = \mathbb{E}_{\hat{\mathbf{x}}\sim\mathcal{D}_s}[p(y|\hat{\mathbf{x}})]. \tag{12}$$

Higher is better.

**PSNR.** For paired evaluation, Peak Signal-to-Noise Ratio is computed from the mean squared error (MSE):

$$\text{MSE}(\mathbf{x}, \hat{\mathbf{x}}) = \frac{1}{|\Omega|}\sum_{u\in\Omega}\left(\mathbf{x}_u - \hat{\mathbf{x}}_u\right)^2, \qquad \text{PSNR}(\mathbf{x}, \hat{\mathbf{x}}) = 10\log_{10}\frac{\text{MAX}^2}{\text{MSE}(\mathbf{x}, \hat{\mathbf{x}})}, \tag{13}$$

where $\text{MAX}$ is the maximum possible pixel value after normalization (e.g., $\text{MAX} = 1$). Higher is better.

**SSIM.** Structural Similarity (SSIM) compares local luminance, contrast, and structure between $\mathbf{x}$ and $\hat{\mathbf{x}}$:

$$\text{SSIM}(\mathbf{x}, \hat{\mathbf{x}}) = \frac{(2\mu_x\mu_{\hat{x}} + c_1)(2\sigma_{x\hat{x}} + c_2)}{(\mu_x^2 + \mu_{\hat{x}}^2 + c_1)(\sigma_x^2 + \sigma_{\hat{x}}^2 + c_2)}, \tag{14}$$

where $\mu$ and $\sigma$ denote local means and standard deviations, and $\sigma_{x\hat{x}}$ is the local covariance; $c_1, c_2$ are small constants for numerical stability. Higher is better.

**LPIPS.** We report the Learned Perceptual Image Patch Similarity (LPIPS), which measures perceptual distance using deep features. Given a pretrained network with layer features $\phi_\ell(\cdot)$, LPIPS is

$$\text{LPIPS}(\mathbf{x}, \hat{\mathbf{x}}) = \sum_\ell w_\ell \left\|\frac{\phi_\ell(\mathbf{x})}{\|\phi_\ell(\mathbf{x})\|_2} - \frac{\phi_\ell(\hat{\mathbf{x}})}{\|\phi_\ell(\hat{\mathbf{x}})\|_2}\right\|_2^2, \tag{15}$$

with fixed learned weights $w_\ell$. Lower is better.

**Dice.** The Dice coefficient (a.k.a. F1 score on pixels) between predicted mask $A$ and ground-truth mask $B$ is

$$\text{Dice}(A, B) = \frac{2|A \cap B|}{|A| + |B|} = \frac{2\,\text{TP}}{2\,\text{TP} + \text{FP} + \text{FN}}. \tag{16}$$

Higher is better.

**mIoU.** Intersection-over-Union is

$$\text{IoU}(A, B) = \frac{|A \cap B|}{|A \cup B|} = \frac{\text{TP}}{\text{TP} + \text{FP} + \text{FN}}, \tag{17}$$

and mean IoU (mIoU) averages IoU over classes (binary foreground/background in our setting). Higher is better.

**Box matching and IoU.** A predicted box is matched to a ground-truth box if its IoU exceeds a threshold $\tau$ and it is the highest-IoU unmatched prediction for that ground truth. IoU is defined as intersection area over union area between the two boxes.

*Table 9.* Image fidelity of edited CAG measured against the original images. Higher PSNR/SSIM and lower LPIPS indicate the edited result remains more similar to the input, reflecting higher visual realism and better structure preservation.

| Method | PSNR↑ | LPIPS↓ | SSIM↑ |
|---|---|---|---|
| Pix2PixHD (Wang et al., 2018) | 17.807 | 0.704 | 0.676 |
| SPADE (Park et al., 2019) | 14.951 | 0.600 | 0.577 |
| SDEdit (Meng et al., 2021) | 16.820 | 0.410 | 0.705 |
| SDM (Wang et al., 2022) | 14.199 | 0.485 | 0.616 |
| SiameseDiff (Qiu et al., 2025a) | 18.230 | 0.281 | 0.790 |
| **OT-Bridge Editor (ours)** | **20.983** | **0.248** | **0.878** |

**Precision, Recall, and F1.** At a given confidence threshold, we compute

$$\text{Precision} = \frac{\text{TP}}{\text{TP} + \text{FP}}, \qquad \text{Recall} = \frac{\text{TP}}{\text{TP} + \text{FN}}, \qquad \text{F1} = \frac{2 \cdot \text{Precision} \cdot \text{Recall}}{\text{Precision} + \text{Recall}}. \tag{18}$$

Higher is better.

**mAP@0.5 and mAP@0.75.** We report mean Average Precision (mAP) at IoU thresholds $\tau = 0.5$ and $\tau = 0.75$. For each $\tau$, AP is computed as the area under the precision–recall curve obtained by sweeping the confidence threshold over all predictions, and mAP averages AP over classes (single-class stenosis detection in our setting). Higher is better.

**Editable condition construction.** For each angiography image $x$, we obtain a vessel segmentation mask $m$ and compute an edge map $e = \mathcal{E}(x)$. The stacked 3-channel condition is

$$y = \text{Stack}\big(m, \; e \odot m, \; \partial m\big) \in \mathbb{R}^{H \times W \times 3}. \tag{19}$$

To create stenosis edits, we modify $m$ into $\tilde{m}$ by imposing a target stenosis degree at a specified vessel location (proximal/mid/distal), and form $\tilde{y} = \text{Stack}(\tilde{m}, \; e \odot \tilde{m}, \; \partial \tilde{m})$. All methods are evaluated under the same edited conditions $\tilde{y}$.

# B. Additional Results

*Table 10.* Detection performance on ARCADE and the internal dataset. All results are evaluated on real Val/Test splits. We report mean±std over 3 runs.

| Dataset | Detector | Training | mAP@0.5↑ | mAP@0.75↑ | Precision↑ | Recall↑ | F1↑ |
|---------|----------|----------|----------|-----------|------------|---------|-----|
| **ARCADE** | | | | | | | |
| | YOLOv8 | Real-only | 0.525±0.009 | 0.312±0.011 | 0.740±0.010 | 0.603±0.012 | 0.664±0.009 |
| | | Synth-only | 0.662±0.008 | 0.401±0.010 | 0.768±0.009 | 0.709±0.011 | 0.737±0.008 |
| | | Real+Synth | 0.727±0.006 | 0.465±0.009 | 0.812±0.008 | 0.741±0.010 | 0.775±0.007 |
| | DINO-DETR | Real-only | 0.508±0.010 | 0.300±0.012 | 0.712±0.012 | 0.589±0.013 | 0.645±0.010 |
| | | Synth-only | 0.615±0.012 | 0.360±0.013 | 0.732±0.011 | 0.666±0.014 | 0.697±0.011 |
| | | Real+Synth | 0.720±0.007 | 0.452±0.010 | 0.804±0.009 | 0.732±0.011 | 0.766±0.008 |
| | Grounding DINO | Real-only | 0.276±0.012 | 0.141±0.010 | 0.521±0.015 | 0.402±0.018 | 0.453±0.014 |
| | | Synth-only | 0.330±0.015 | 0.170±0.012 | 0.548±0.016 | 0.468±0.017 | 0.505±0.015 |
| | | Real+Synth | 0.418±0.011 | 0.232±0.012 | 0.612±0.014 | 0.523±0.016 | 0.564±0.013 |
| | RTMDet | Real-only | 0.545±0.008 | 0.325±0.010 | 0.756±0.010 | 0.630±0.012 | 0.687±0.009 |
| | | Synth-only | 0.625±0.010 | 0.380±0.012 | 0.778±0.009 | 0.681±0.013 | 0.726±0.010 |
| | | Real+Synth | 0.675±0.007 | 0.420±0.010 | 0.802±0.009 | 0.703±0.011 | 0.749±0.008 |
| **Multi-center Internal** | | | | | | | |
| | YOLOv8 | Real-only | 0.654±0.011 | 0.395±0.013 | 0.805±0.012 | 0.660±0.014 | 0.725±0.010 |
| | | Synth-only | 0.582±0.014 | 0.348±0.015 | 0.750±0.014 | 0.572±0.016 | 0.648±0.013 |
| | | Real+Synth | 0.731±0.008 | 0.442±0.010 | 0.865±0.010 | 0.708±0.011 | 0.779±0.007 |
| | DINO-DETR | Real-only | 0.638±0.010 | 0.380±0.012 | 0.782±0.011 | 0.650±0.012 | 0.710±0.009 |
| | | Synth-only | 0.565±0.013 | 0.335±0.014 | 0.725±0.013 | 0.565±0.015 | 0.635±0.012 |
| | | Real+Synth | 0.725±0.007 | 0.435±0.009 | 0.850±0.008 | 0.702±0.010 | 0.768±0.008 |
| | Grounding DINO | Real-only | 0.385±0.012 | 0.215±0.013 | 0.605±0.015 | 0.475±0.016 | 0.532±0.014 |
| | | Synth-only | 0.312±0.016 | 0.165±0.015 | 0.552±0.017 | 0.432±0.018 | 0.485±0.015 |
| | | Real+Synth | 0.442±0.010 | 0.255±0.012 | 0.665±0.012 | 0.528±0.014 | 0.588±0.011 |
| | RTMDet | Real-only | 0.615±0.009 | 0.372±0.011 | 0.775±0.010 | 0.630±0.011 | 0.695±0.010 |
| | | Synth-only | 0.548±0.012 | 0.325±0.013 | 0.718±0.012 | 0.550±0.014 | 0.622±0.011 |
| | | Real+Synth | 0.688±0.006 | 0.415±0.008 | 0.835±0.009 | 0.678±0.010 | 0.754±0.007 |

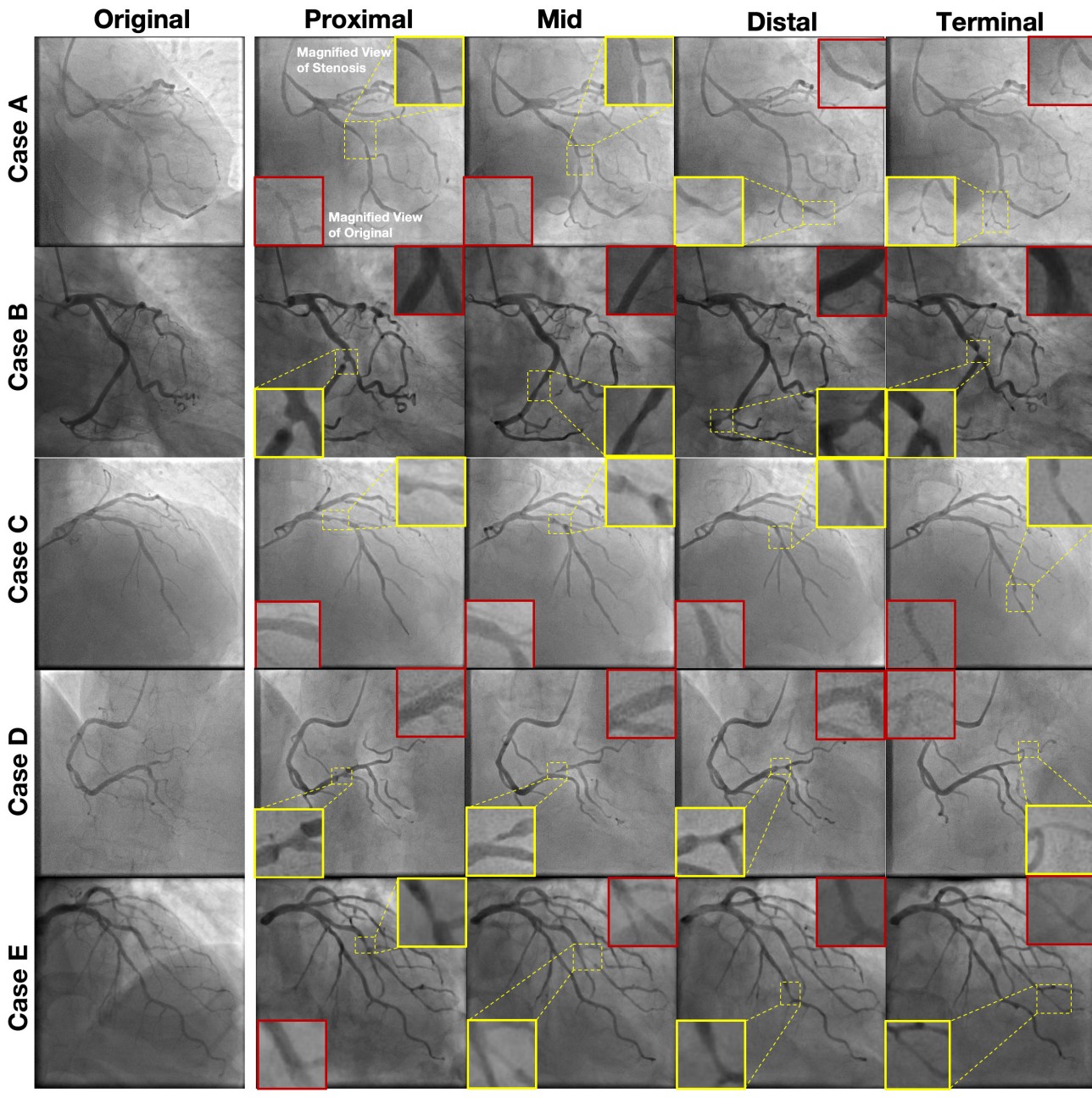

*Figure 9.* Qualitative results for location-controlled stenosis editing across coronary segments. For five representative cases (A-E), we show the original CAG (left) and the edited outputs with target stenosis placed in the Proximal, Mid, Distal, and Terminal segments (columns 2-5). Red boxes mark the target region on the original image, and yellow boxes highlight the same region in the edited results with zoomed-in ROIs for detailed inspection.

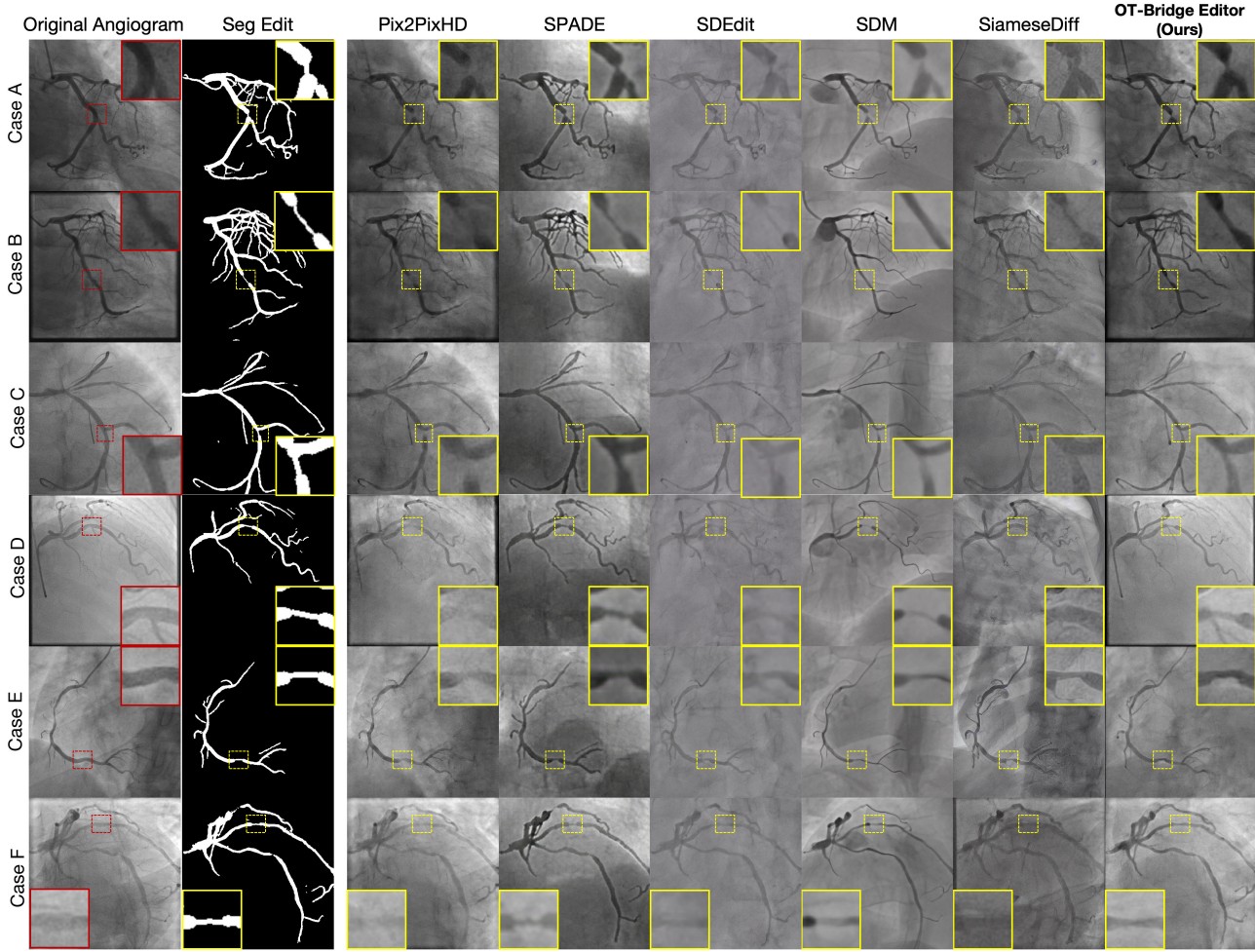

*Figure 10.* Qualitative comparison of geometry-constrained stenosis editing on CAG. For six representative cases (A-F), we show the input CAG and the desired structural change specified by an edited vessel mask, followed by results from baselines and our OT-Bridge Editor. Red boxes indicate the target region on the original image; yellow boxes highlight the same region across methods.

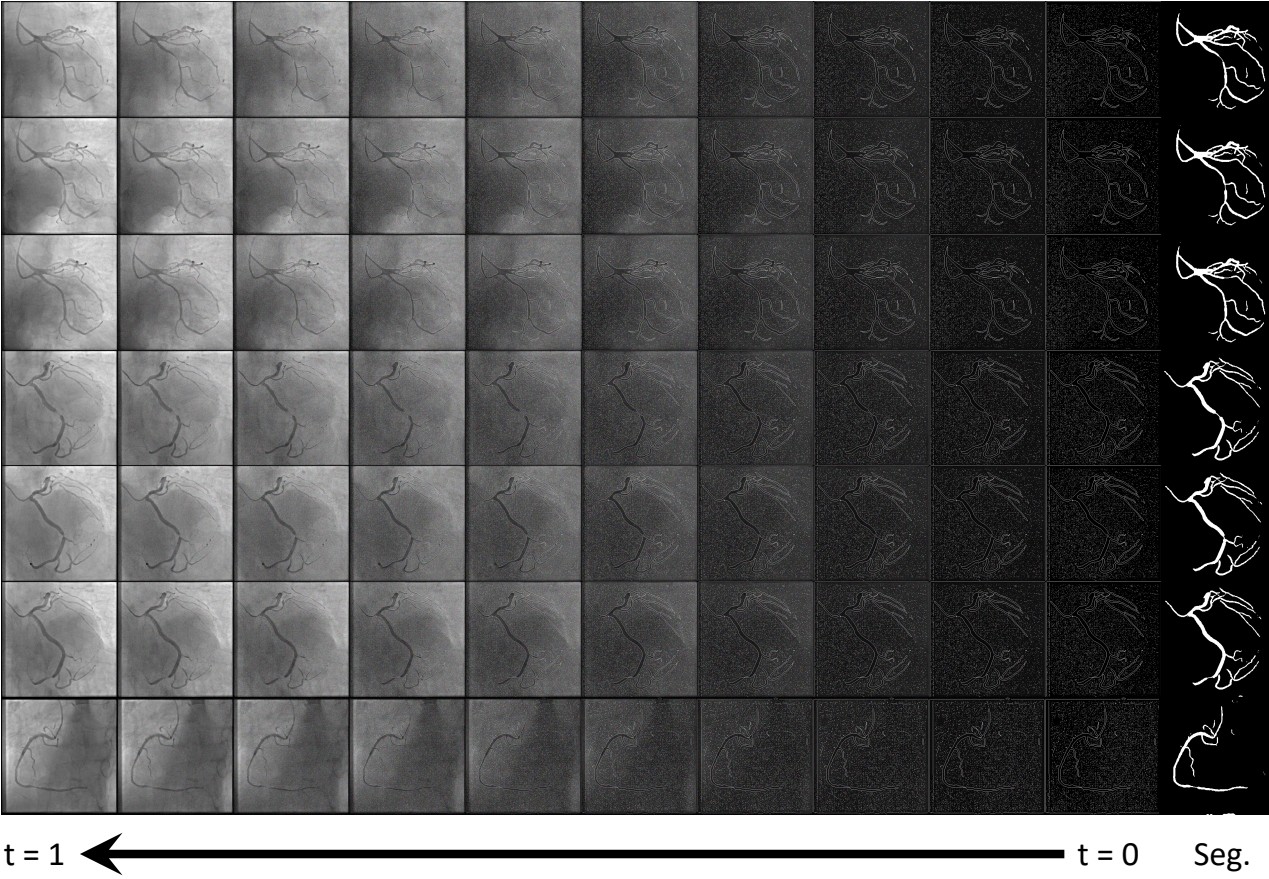

t = 1 ⟵ t = 0   Seg.

*Figure 11.* Visualization of the reverse bridge trajectory under geometric guidance. Columns show intermediate states from $t=0$ to $t=1$ (right to left), starting from the vessel-structure condition; the last column shows the corresponding segmentation constraint. Across diverse cases (rows), the trajectory progressively transfers structural information into the angiogram domain while preserving vessel topology and suppressing off-target drift.

*Table 11.* Impact of different synthesis methods on downstream detection under *Synth-only* and *Real+Synth* (1:1) training. We report mAP@0.5 and F1 on real test splits.

| Synthesis | Training | YOLOv8 | | DINO-DETR | | RTMDet | | Grounding DINO | |
|---|---|---|---|---|---|---|---|---|---|
| | | mAP@0.5 | F1 | mAP@0.5 | F1 | mAP@0.5 | F1 | mAP@0.5 | F1 |
| **ARCADE** | | | | | | | | | |
| Pix2PixHD | Synth-only | 0.137±0.014 | 0.312±0.021 | 0.125±0.016 | 0.305±0.022 | 0.152±0.015 | 0.325±0.020 | 0.082±0.012 | 0.215±0.018 |
| | Real+Synth (1:1) | 0.405±0.013 | 0.550±0.015 | 0.392±0.014 | 0.540±0.016 | 0.420±0.012 | 0.562±0.015 | 0.285±0.013 | 0.405±0.015 |
| SPADE | Synth-only | 0.251±0.016 | 0.402±0.019 | 0.238±0.017 | 0.395±0.020 | 0.275±0.016 | 0.420±0.019 | 0.125±0.015 | 0.268±0.017 |
| | Real+Synth (1:1) | 0.460±0.012 | 0.590±0.014 | 0.445±0.013 | 0.580±0.015 | 0.475±0.012 | 0.600±0.014 | 0.315±0.011 | 0.435±0.013 |
| SiameseDiff | Synth-only | 0.183±0.015 | 0.355±0.020 | 0.170±0.016 | 0.340±0.021 | 0.205±0.015 | 0.370±0.020 | 0.098±0.014 | 0.242±0.019 |
| | Real+Synth (1:1) | 0.445±0.012 | 0.575±0.014 | 0.430±0.013 | 0.565±0.015 | 0.460±0.011 | 0.588±0.014 | 0.302±0.012 | 0.422±0.014 |
| SDEdit | Synth-only | 0.210±0.017 | 0.372±0.018 | 0.195±0.018 | 0.360±0.019 | 0.225±0.017 | 0.385±0.018 | 0.110±0.016 | 0.255±0.018 |
| | Real+Synth (1:1) | 0.452±0.011 | 0.582±0.013 | 0.438±0.012 | 0.572±0.014 | 0.465±0.011 | 0.592±0.013 | 0.308±0.010 | 0.428±0.012 |
| SDM | Synth-only | 0.195±0.016 | 0.365±0.019 | 0.182±0.017 | 0.352±0.020 | 0.212±0.016 | 0.378±0.019 | 0.105±0.015 | 0.248±0.018 |
| | Real+Synth (1:1) | 0.440±0.012 | 0.570±0.014 | 0.425±0.013 | 0.560±0.015 | 0.455±0.011 | 0.584±0.014 | 0.295±0.011 | 0.415±0.013 |
| DiGDA | Synth-only | – | – | – | – | – | – | – | – |
| | Real+Synth (1:1) | 0.484 | 0.519 | – | – | – | – | – | – |
| **OT-Bridge Editor (Ours)** | Synth-only | 0.647±0.009 | 0.640±0.011 | 0.520±0.010 | 0.570±0.012 | 0.575±0.011 | 0.545±0.012 | 0.325±0.012 | 0.453±0.014 |
| | Real+Synth (1:1) | 0.690±0.009 | 0.720±0.010 | 0.670±0.010 | 0.710±0.011 | 0.705±0.009 | 0.730±0.010 | 0.450±0.009 | 0.564±0.010 |
| **Multi-center Internal** | | | | | | | | | |
| Pix2PixHD | Synth-only | 0.254±0.015 | 0.412±0.018 | 0.225±0.017 | 0.395±0.019 | 0.245±0.016 | 0.408±0.018 | 0.155±0.016 | 0.320±0.019 |
| | Real+Synth (1:1) | 0.658±0.012 | 0.728±0.011 | 0.642±0.013 | 0.712±0.010 | 0.620±0.011 | 0.698±0.011 | 0.390±0.014 | 0.538±0.016 |
| SPADE | Synth-only | 0.352±0.014 | 0.485±0.016 | 0.315±0.015 | 0.460±0.017 | 0.338±0.015 | 0.472±0.016 | 0.205±0.015 | 0.385±0.018 |
| | Real+Synth (1:1) | 0.665±0.011 | 0.732±0.010 | 0.650±0.012 | 0.718±0.010 | 0.632±0.010 | 0.705±0.009 | 0.405±0.013 | 0.550±0.015 |
| SiameseDiff | Synth-only | 0.415±0.013 | 0.545±0.015 | 0.390±0.014 | 0.520±0.016 | 0.400±0.014 | 0.535±0.015 | 0.245±0.014 | 0.415±0.016 |
| | Real+Synth (1:1) | 0.682±0.010 | 0.745±0.009 | 0.665±0.011 | 0.730±0.009 | 0.645±0.009 | 0.718±0.010 | 0.420±0.012 | 0.562±0.013 |
| SDEdit | Synth-only | 0.380±0.016 | 0.510±0.017 | 0.365±0.018 | 0.495±0.019 | 0.370±0.017 | 0.505±0.018 | 0.225±0.017 | 0.400±0.018 |
| | Real+Synth (1:1) | 0.675±0.011 | 0.738±0.010 | 0.658±0.012 | 0.724±0.010 | 0.638±0.011 | 0.710±0.011 | 0.412±0.012 | 0.558±0.013 |
| SDM | Synth-only | 0.360±0.015 | 0.492±0.016 | 0.345±0.017 | 0.480±0.018 | 0.350±0.016 | 0.490±0.017 | 0.218±0.016 | 0.395±0.017 |
| | Real+Synth (1:1) | 0.670±0.012 | 0.735±0.011 | 0.652±0.013 | 0.720±0.011 | 0.635±0.011 | 0.708±0.011 | 0.408±0.012 | 0.555±0.013 |
| **OT-Bridge Editor (Ours)** | Synth-only | 0.582±0.014 | 0.648±0.013 | 0.565±0.013 | 0.635±0.012 | 0.548±0.012 | 0.622±0.011 | 0.312±0.016 | 0.485±0.015 |
| | Real+Synth (1:1) | 0.731±0.008 | 0.779±0.007 | 0.725±0.007 | 0.768±0.008 | 0.688±0.006 | 0.754±0.007 | 0.442±0.010 | 0.588±0.011 |

*Table 12.* Scaling synthetic augmentation on ARCADE with OT-Bridge Editor. We vary the synthetic ratio $r$ while keeping the real training set fixed ($N_{\text{synth}} = r\, N_{\text{real}}$). Rec@P=$\tau$ denotes recall at a fixed precision of 0.90.

| $r$ | mAP@0.5↑ | F1↑ | Rec@P=$\tau$ ↑ |
|---|---|---|---|
| 0 | $0.525_{\pm0.009}$ | $0.664_{\pm0.009}$ | $0.485_{\pm0.011}$ |
| 0.1 | $0.595_{\pm0.010}$ | $0.705_{\pm0.010}$ | $0.545_{\pm0.012}$ |
| 0.25 | $0.655_{\pm0.008}$ | $0.738_{\pm0.009}$ | $0.595_{\pm0.010}$ |
| 0.5 | $0.698_{\pm0.007}$ | $0.760_{\pm0.008}$ | $0.635_{\pm0.009}$ |
| 1.0 | $\mathbf{0.727}_{\pm0.006}$ | $\mathbf{0.775}_{\pm0.007}$ | $\mathbf{0.668}_{\pm0.008}$ |
| 2.0 | $0.725_{\pm0.006}$ | $0.771_{\pm0.006}$ | $0.665_{\pm0.008}$ |

