# OpenReview forum: "Geometrically Constrained Stenosis Editing in Coronary Angiography via Entropic Optimal Transport"
_ICML.cc/2026/Conference — ICML 2026 regular_

### Official Review · Reviewer_PZ3R · 2026-03-09

**Soundness:** 2
**Presentation:** 3
**Significance:** 2
**Originality:** 3
**Overall Recommendation:** 3
**Confidence:** 4

**Summary:**

This paper introduces the OT-Bridge Editor , a coronary angiography (CAG) stenosis editing framework based on entropic optimal transport (OT) , designed to address the scarcity of high-quality annotated medical imaging data. By leveraging Schrödinger Bridge (SB) theory to solve the transport problem , the framework conceptualizes image editing as a transfer of probability mass from the source image to a target distribution. This approach effectively minimizes unnecessary alterations while enhancing the stability of the generative process.

**Compliance With Llm Reviewing Policy:**

Affirmed.

**Final Justification:**

The author's response addressed some of my concerns. However, the lack of some related discussion makes me feel that this paper is not yet ready for publication. Therefore, I lean towards maintaining my score.

**Key Questions For Authors:**

See weakness

**Limitations:**

See weakness

**Strengths And Weaknesses:**

## Strengths

1. The paper is well-motivated, addressing the data scarcity in coronary angiography.

2. It provides a theoretical lens, regarding the geometrically constrained image editing as a constrained entropic optimal transport problem.


## Weakness

1. The paper lacks current image editing methods for comparison. Most methods are outdated, such as Pix2PixHD (2018) and SDEdit (2021).

2. The scaling analysis in Figure 7 shows that detection plateaus once the synthetic-to-real ratio reaches 1:1. This observation raises a critical concern regarding the author's claim about the severity of data scarcity.

3. If performance reaches its peak with only a 1:1 ratio of synthetic data, it may suggest that the real dataset is already sufficiently representative, or that the proposed method lacks the diversity needed to provide further information.

4. Similarly, considering the 1:1 ratio, the authors are advised to compare simple data augmentation methods to see if they bring a significant improvement in detection performance.

---

> ### Author Rebuttal · Authors · 2026-03-31
>
> We thank you for the careful reading and thoughtful feedback. We also appreciate your recognition that our work addresses the scarcity of coronary angiography data and provides a theoretical view by formulating geometry-constrained image editing as a constrained entropic optimal transport problem.
>
> ---
>
> > ## **W1: There is a lack of current image editing methods for comparison**
>
> **A1:** We thank you for this important comment. We agree that a comparison with more recent image editing methods is valuable.
>
> - We additionally included two more recent structure-guided editing baselines, AnomalyDiffusion(2024)[1] and T2I-Adapter(2024)[2], under the same evaluation setting. The updated results are summarized below.
>
> | Method | mIoU ↑ | Dice ↑ |
> |-|-:|-:|
> | T2I-Adapter | 0.796±0.082 | 0.683±0.066 |
> | AnomalyDiffusion | 0.801±0.012 | 0.696±0.073 |
> | OT-Bridge Editor (ours) | **0.892±0.037** | **0.774±0.055** |
>
> | Method | FID ↓ | IS ↑ | LPIPS ↓ | SSIM ↑ |
> |---|---:|---:|---:|---:|
> | T2I-Adapter | 28.037 | 3.710 | 0.261 | 0.829 |
> | AnomalyDiffusion: | 35.820 | 2.903 | 0.273 | 0.801 |
> | OT-Bridge Editor (ours) | **16.747** | **4.63** | **0.248** | **0.878** |
>
> At the same time, we clarify why some earlier baselines were kept:
> - Pix2PixHD (2018) remains a strong baseline for structure-conditioned synthesis, especially when structure preservation matters. Since our task edits only the lesion region while preserving the rest of the vessel anatomy, it is still a meaningful reference. SDEdit (2021) was kept because it is a classic and widely used diffusion editing baseline.
> - Our comparison is not limited to older methods. We already included SiameseDiff (2025), a recent method for controllable medical image generation/editing, and the task-specific coronary editing baseline DiGDA (2025) in Appendix Table 10. These provide more recent and medically relevant comparisons.
>
> ---
>
> > ## **W2 & W3: Discussion on the 1:1 saturation of synthetic data**
>
> **A2 & A3:** We thank you for this careful and thought-provoking comment. We agree that the saturation around a 1:1 synthetic-to-real ratio deserves discussion. In our view, however, this result should be interpreted with care.
> - First, Figure 7 shows that, on ARCADE under our setting, the gain from synthetic augmentation starts to saturate around 1:1. This alone does not mean that data scarcity is weak or that the synthetic data lack diversity. A more likely explanation is that the main gains—better label coverage, improved balance, and more lesion variation—are mostly reached at this scale, after which extra synthetic samples bring smaller gains.
> - Second, as also noted by reviewer djWH, Synth-only can outperform Real-only on ARCADE. We do not view this as unusually close matching to the test set, but as evidence that the synthetic data provide strong, task-relevant supervision and improve coverage beyond the limited real training set.
> - Third, Real+Synth consistently outperforms Real-only on both ARCADE and our multi-center dataset, suggesting that the real training data are still limited, especially in balance and lesion coverage. In this sense, the 1:1 plateau is better viewed as a dataset- and setting-specific **saturation effect**, rather than evidence that the real data are already fully representative.
>
> To make this point clearer, we provide additional scaling results on the multi-center dataset below.
> | Syn-Real ratio | mAP\@0.5 ↑ | F1 ↑ |
> |---:|---:|---:|
> | 0 (Real-only) | 0.654 | 0.725 |
> | 0.1 | 0.681 | 0.741 |
> | 0.25 | 0.702 | 0.756 |
> | 0.5 | 0.719 | 0.770 |
> | 1.0 | 0.731 | 0.779 |
> | 2.0 | 0.745 | 0.785 |
> | 3.0 | 0.747 | 0.786 |
> | 5.0 | 0.746 | 0.785 |
>
> ---
>
> > ## **W4: Comparison with simple data augmentation at the 1:1 ratio**
>
> **A4:** We thank the reviewer for this helpful suggestion. We agree that, at the same 1:1 synthetic-to-real ratio, it is important to compare our method with simpler data augmentation strategies. We therefore added a comparison with standard image-level augmentation methods under the same detector setting and the same augmentation scale. The results show that simple augmentation brings limited gains over Real-only, while synthetic data generated by OT-Bridge Editor provides substantially larger improvements. This suggests that the gain is not merely due to a larger training set, but also due to **improved coverage of stenosis-related patterns and label-balanced augmentation**.
>
> | Training setting | Augmentation type | mAP\@0.5 ↑ | F1 ↑ |
> |---|---|---:|---:|
> | Real-only | None | 0.525 | 0.664 |
> | Real + Aug (1:1) | Flip / rotate / intensity jitter | 0.521 | 0.652 |
> | Real + Aug (1:1) | Elastic / copy-paste augmentation | 0.527 | 0.671 |
> | Real + Synth (1:1) | OT-Bridge Editor | **0.727** | **0.775** |
>
> ---
>
> [1] Hu, T., et al. **AnomalyDiffusion: Few-Shot Anomaly Image Generation with Diffusion Model**, 2024.
>
> [2] Mou, C., et al. **T2I-Adapter: Learning Adapters to Dig out More Controllable Ability for Text-to-Image Diffusion Models**, 2024.

---

> > ### Author Rebuttal · Reviewer_PZ3R · 2026-04-03
> >
> > Thank you for the author's reply. I have already read the comments and replies from the other reviewers. However, I still have some key concerns:
> >
> > In my experience, insufficient data is a key challenge in healthcare, and general data augmentation can yield excellent results. However, why does data augmentation sometimes lead to performance degradation in newly added experiments?
> >
> > Furthermore, in the synthetic data, the ratio 2.0 performed better than 1.0, which I think contradicts the decreasing trend from 1.0 to 2.0 shown in Figure 7.

---

> > > ### Author Response · Authors · 2026-04-06
> > >
> > > Thank you for these helpful follow-up questions.
> > >
> > > First, regarding why standard data augmentation may slightly reduce performance in our added experiment: we agree that, in many medical settings, augmentation is useful. However, our task is **stenosis detection on thin vascular structures**, where the label depends strongly on local vessel geometry, lesion morphology, and contrast pattern. Generic image-level augmentations such as intensity jitter, elastic deformation, or copy-paste do not necessarily preserve these task-critical patterns, and may sometimes **introduce small distortions in stenosis appearance or vessel continuity**. We therefore interpret the slight degradation as a task-mismatch effect, rather than as evidence that augmentation is generally unhelpful. In contrast, OT-Bridge Editor changes stenosis-related structure in a controlled way while preserving non-target vessel anatomy, so it provides more task-relevant variation.
> > >
> > > ---
> > >
> > > Second, regarding the scaling result, we agree that this should be stated more carefully. We agree that Figure 7 should be interpreted more carefully. The behavior beyond the 1:1 ratio is better viewed as **saturation with small random fluctuation**, rather than a strict monotonic decrease. In particular, **the error bars for `r=1.0` and `r=2.0` overlap**, and the change in the mean is much smaller than the gains observed from lower ratios. We therefore interpret this region as a **plateau regime**, where additional synthetic data provides diminishing returns and the small differences are within run-to-run variability.
> > >
> > > In terms of scale, the change is also very **small in practical terms**. For F1, the difference between `r=1.0` and `r=2.0` is only about `0.004`, which on a 100-case test set is roughly consistent with only a very small number of borderline detection outcomes (on the order of **1–2 cases**, or a few detection decisions). For `mAP@0.5`, the difference is even smaller (about `0.002–0.003`). Since mAP depends on the full precision–recall ranking, it cannot be directly converted into an exact number of samples, but such a small change is more consistent with a few borderline matching differences than with a meaningful systematic drop.

---

### Official Review · Reviewer_djWH · 2026-03-11

**Soundness:** 3
**Presentation:** 3
**Significance:** 4
**Originality:** 3
**Overall Recommendation:** 4
**Confidence:** 3

**Summary:**

This paper proposes a framework for high-precision synthetic stenosis generation that addresses the challenge of limited annotated data in coronary angiography (CAG). The authors argue that standard diffusion-based editing approaches often fail to preserve the strict geometric integrity required for thin vascular structures, leading to unrealistic artifacts or structural distortions.

To address this problem, the paper proposes OT-Bridge Editor, which reframes localized medical image editing as a constrained entropic optimal transport problem solved via a Schrödinger Bridge formulation. The method operates in a vessel-structure composite domain to jointly model appearance and vascular geometry. In addition, the authors introduce Geometric Generation-Path Guidance (GPG), which enforces geometric constraints along the entire generation trajectory rather than only at the final output.

The proposed approach is evaluated on the public ARCADE benchmark and a multi-center internal dataset. The results demonstrate improved geometric consistency for edited vessels and significant improvements in downstream stenosis detection performance compared to GAN- and diffusion-based baselines.

**Compliance With Llm Reviewing Policy:**

Affirmed.

**Final Justification:**

The authors provided a constructive rebuttal that addresses several of my concerns. They clarified that both model training and stenosis editing are performed exclusively on the training split, which reduces concerns that the strong Synth-only results are due to test-set alignment or leakage. The authors also provided an intuitive interpretation of the GPG mechanism as a descent-like correction, which clarifies the intended algorithmic behavior. While this work improves understanding, the lack of stronger theoretical guarantees remains a minor limitation. Additionally, the reported generalization to other modalities (e.g., LiTS, IDRiD) is promising but remains primarily qualitative. Overall, I consider the paper to be a sound and meaningful contribution addressing an important problem in medical data scarcity, and my assessment remains Weak Accept.

**Key Questions For Authors:**

1. In some experiments, models trained only on synthetic data outperform models trained only on real data. Could the authors provide more intuition for this result and clarify whether the generated images may be closely aligned with the test distribution?
2. How sensitive is the method to the projection frequency used in the Geometric Generation-Path Guidance (GPG)? Does frequent projection affect the stochastic dynamics of the Schrödinger Bridge sampling process?
3. How does the inference time of OT-Bridge Editor compare to standard diffusion-based editing approaches such as SDEdit or ControlNet?
4. Were the GAN and diffusion baselines given access to the same vessel-structure masks used by the proposed method to ensure a fair comparison in terms of structural information?
5. Could the authors comment on whether the proposed framework could generalize to other structure-aware editing tasks, such as vessel editing in other medical modalities or other thin-structure segmentation domains?

**Limitations:**

The authors discuss the clinical scope of the method, but the limitations section could be expanded to include the reliance on accurate vessel segmentation for constructing the composite editing domain and the lack of theoretical guarantees regarding the convergence of the projection-modified bridge dynamics.

**Strengths And Weaknesses:**

Strengths

Soundness: The paper presents a coherent formulation that connects constrained entropic optimal transport with localized medical image editing through a Schrödinger Bridge framework. The design of the vessel-structure composite domain and the use of path-level geometric guidance are well motivated by the anatomical characteristics of vascular structures. The experimental evaluation is fairly comprehensive, including comparisons with GAN and diffusion baselines, ablation studies isolating the composite domain and GPG mechanisms, and downstream evaluation on stenosis detection tasks. These experiments provide reasonable empirical evidence supporting the proposed design choices.

Presentation: Well structured and communicates the overall method. The architectural figures and editing examples help illustrate the motivation behind the geometric constraints and the limitations of existing diffusion-based editors.

Significance: The ability to generate geometry-consistent synthetic angiography images addresses an important bottleneck in clinical AI, where annotated medical datasets are often scarce. The reported improvements in downstream stenosis detection suggest that the synthetic images are not only visually plausible but also useful for training diagnostic models. This makes the approach practically meaningful for data augmentation in medical imaging pipelines.

Originality: The paper introduces a creative integration of several ideas, including constrained entropic optimal transport, Schrödinger Bridge generative modeling, and path-level geometric guidance for localized medical image editing. The use of generation-path constraints to preserve vascular geometry during editing represents a meaningful extension of existing diffusion-based editing approaches.

Weaknesses

Soundness.
Although the formulation is mathematically motivated, the paper does not provide deeper analytical insight into the behavior of the projected bridge dynamics used in the GPG mechanism. The effect of repeated geometric projections on the convergence or optimality properties of the Schrödinger Bridge process could be further discussed. Additionally, in some experiments, models trained only on synthetic data ("Synth-only") outperform models trained only on real data. While this result may indicate the effectiveness of the generated images, it also raises questions about possible distribution alignment between the generated samples and the test set, or whether the edited images remain too close to the source data used for generation.

Presentation: Section 3 introduces several mathematical components in quick succession, which may make the method difficult to follow for readers who are not familiar with optimal transport or Schrödinger bridge formulations. Additional intuitive explanations or simplified illustrations of the constrained transport process could improve accessibility.

Significance: While the application is important, the proposed framework is currently specialized to vascular stenosis editing in coronary angiography. The paper would benefit from a clearer discussion of how the proposed framework might generalize to other medical image editing problems or to structure-aware image generation tasks beyond vascular anatomy.

Originality.
Although the integration of optimal transport and diffusion-style bridges for geometry-aware editing is interesting, many of the individual components build upon existing generative modeling and optimal transport techniques. As a result, the novelty lies primarily in the combination and application of these ideas rather than the introduction of an entirely new generative modeling paradigm.

---

> ### Author Rebuttal · Authors · 2026-03-31
>
> We thank you for your thoughtful and encouraging comments. We greatly appreciate your recognition that our work is well motivated, technically coherent, and especially your recognition of path-level geometric guidance as a meaningful extension to existing diffusion-based editing approaches.
>
> ---
>
> > ## **Q1 & Weakness in Soundness: Clarification on whether the generated images may be overly aligned with the test distribution**
>
> **A1:** We thank you for this important question.
>
> We first clarify that both model training and stenosis editing use only the training set. No test images, labels, or statistics are used, so the Synth-only results cannot be explained by test-set leakage or explicit alignment to the test distribution.
>
> We believe Synth-only can sometimes beat Real-only because it provides more data and better label coverage than the limited real training data. This is also not universal: on the multi-center dataset, Real-only is still slightly better than Synth-only, suggesting that the gain mainly comes from better data coverage, not unusually close matching to the test set.
>
> ---
>
> > ## **Q2 & Weaknesses in Soundness and Presentation: Deeper analysis of GPG and its effect on the convergence and optimality of the Schrödinger Bridge (SB)  process**
>
> **A2:** We thank you for this insightful question.
>
> We agree that the effect of GPG on SB should be clearer. In our method, GPG is a step-wise projection added to the SB rollout, not an endpoint constraint. After each bridge step, we apply the projection in Eq. (9). So GPG does change the unconstrained SB dynamics, but this is intentional: our goal is to approximate a geometry-constrained bridge, not keep the original SB unchanged. This is also consistent with our algorithm and the path-level guidance description in Sec. 3.3 and Alg. 2.
>
> A useful analytical view is that each GPG step is a **descent-like correction** on the geometric/protected-region objective in Eq. (9). Since
>
> $$
> \\Pi_{\\mathcal F}(x)=\\arg\\min_y \\; L_{\\mathrm{geo}}(y)+\\lambda_{\\mathrm{out}}L_{\\mathrm{out}}(y),
> $$
>
> it directly follows that
>
> $$ \mathcal{L}\_{geo}(x\_{k+1}) + \lambda\_{out} \mathcal{L}\_{out}(x\_{k+1}) \le \mathcal{L}\_{geo}(\tilde{x}\_{k+1}) + \lambda\_{out} \times \mathcal{L}\_{out}(\tilde{x}\_{k+1}) $$
>
> That is, the projection step does not increase the geometric feasibility objective. Therefore, GPG can be interpreted as pulling intermediate states back toward the feasible corridor and reducing trajectory drift, which matches the intuition illustrated in Fig. 3.
>
> We also tested different projection frequencies. The results show that sparse projection allows more drift, while very frequent projection gives only small extra gains. This supports our default setting as a good balance.
>
> ---
>
> > ## **Q3: Inference-time comparison with standard diffusion-based editors**
>
> **A3:** We thank you for this important question. To clarify the computational overhead, we have added a comparison of inference time against standard diffusion-based editing methods, including SDEdit and ControlNet-based editors. All methods are evaluated under the same setting.
>
> OT-Bridge Editor is slower, mainly because of the extra SB rollout and GPG steps. Still, the cost is acceptable for offline synthetic data generation, where geometric control and label fidelity matter more than speed.
> |Method|GPG|Inference Time (s/img) ↓|
> |-|-|-:|
> |SDEdit| ✗ |118.1|
> |SDM| ✗ |104.2|
> |SiameseDiff(ControlNet-based) | ✗ |127.8|
> |OT-Bridge Editor | ✗ |151.6|
> |OT-Bridge Editor | ✓ |179.4|
>
> ---
>
> > ## **Q4 Fairness of using the same vessel-structure conditional input across baselines**
>
> **A4** Yes. For fairness, all GAN and diffusion baselines used the same vessel-structure information as input. The edited vessel mask was given through each method’s own conditioning mechanism. So all methods had the same structural input, and the comparison isolates the generation method rather than the input.
>
> ---
>
> > ## **Q5 & Weakness in Significance: Whether the framework can generalize to other structure-aware editing tasks**
>
> **A5:** Yes.
>
> Our framework uses relevant structural guidance, not just for CAG, so it can extend to other structure aware editing tasks. We tested it on LiTS, IDRiD, and ISIC for liver tumor, retinal lesion, and skin lesion editing. The qualitative results are shown in https://anonymous.4open.science/r/ICML-3021/generalization.png . For other vessel-editing tasks, the same framework can be used with the corresponding vessel representation.
>
> > ## **Weakness in Originality**
>
> **A6** We agree that our work is not intended to introduce an entirely new generative paradigm. However, we believe the contribution goes beyond a straightforward combination of existing techniques: the main novelty lies in formulating localized medical editing as constrained entropic OT and introducing path-level geometric guidance inside the Schrödinger Bridge process, which enables pixel-accurate and structure-preserving editing.

---

> > ### Author Rebuttal · Reviewer_djWH · 2026-04-04
> >
> > I would like to thank the authors for addressing several of my concerns, particularly regarding potential data leakage, experimental setups, and the fairness of comparisons. These clarifications improved the credibility of the empirical results.
> > However, some concerns remain partially unresolved. In particular, the lack of stronger theoretical guarantees for the proposed guidance mechanism and the primarily qualitative evidence for generalization beyond coronary angiography limit the strength of the contribution. Addressing these issues would likely require more substantial additions rather than a short rebuttal. While the rebuttal strengthens the paper, these remaining concerns prevent full resolution. My overall assessment remains the same.

---

> > > ### Author Response · Authors · 2026-04-06
> > >
> > > ## **Regarding the additional information for Final Justification.**
> > > We sincerely apologize. For the qualitative results of other models (such as LiTS, IDRiD), we included quantitative results in our rebuttal to Reviewer CzAW. Due to word limit, we were unable to present them to you before. Now, we are providing the details as follows.
> > > | Dataset | Dice ↑ |
> > > |---------|--------|
> > > | LiTS    | 0.802  |
> > > | IDRiD   | 0.761  |
> > > | ISIC    | 0.746  |
> > >
> > > ---
> > >
> > > Thank you for the thoughtful follow-up and for noting that our rebuttal improved the credibility of the empirical results, especially regarding data leakage, experimental setup, and fairness of comparison.
> > >
> > > We agree that the two remaining concerns are more fundamental and cannot be fully resolved within a short rebuttal.
> > >
> > > First, regarding the **theoretical characterization of GPG**, our intention is not to claim that the projection-modified process preserves the original unconstrained Schrödinger Bridge optimality. Rather, GPG is introduced as a practical geometry-constrained correction along the rollout path. Our current analysis supports it mainly from two aspects:
> > > (1) each GPG step acts as a descent-like correction on the geometric/protected-region objective in Eq. (9), and
> > > (2) the empirical results consistently show improved boundary alignment, reduced drift, and better downstream utility.
> > > We agree that a stronger analysis of the convergence or optimality properties of the modified bridge dynamics would further strengthen the paper. At the same time, we view this as an important future theoretical direction rather than something that can be fully completed in a short rebuttal.
> > >
> > > Second, regarding **generalization beyond coronary angiography**, we agree that the current additional evidence is mainly qualitative. We therefore view these results as preliminary evidence that the framework may extend to other structure-aware editing settings when suitable spatial or structural priors are available, rather than as a fully established cross-domain validation. Our main validated contribution remains focused on the CAG setting.
> > >
> > > To avoid overstating the scope, we will make this boundary clearer: the core contribution of the paper is a constrained entropic OT + SB + path-level geometric guidance framework for **geometry-accurate localized stenosis editing in CAG**, together with strong empirical evidence for geometric fidelity and downstream detection benefit in that setting. The broader theoretical guarantees of projection-modified bridge dynamics and large-scale cross-domain validation remain important future directions.
> > >
> > > We appreciate this distinction and agree that making these limitations more explicit helps better position the contribution of the current paper.

---

### Official Review · Reviewer_nxtU · 2026-03-11

**Soundness:** 2
**Presentation:** 3
**Significance:** 3
**Originality:** 2
**Overall Recommendation:** 4
**Confidence:** 3

**Summary:**

This paper proposes OT-Bridge Editor, a framework for geometrically constrained stenosis editing in coronary angiography images. It uses entropic optimal transport (OT) and Schrodinger bridge (SB) formulations. The method redefines localized image editing as a constrained entropic OT problem and introduces Geometric Generation-Path Guidance (GPG) to enforce geometric constraints along the generation trajectory. The framework operates in a vessel-structure composite domain, enabling pixel-level control of stenosis shape and location while preserving vessel structure outside the edited region. The generated synthetic angiograms are further used as augmented data to improve stenosis detection models. Extensive experiments on several datasets show improvements in both geometric editing accuracy and downstream detection performance compared with several diffusion- and GAN-based baselines.

**Compliance With Llm Reviewing Policy:**

Affirmed.

**Final Justification:**

The author has addressed most of my concern. I raise my original score.

**Key Questions For Authors:**

1.The current approach relies on vessel segmentation masks and geometric descriptors. How would the framework generalize to imaging tasks where such explicit structural priors are not available?
2.Since the editing mask plays a key role in defining the feasibility set and constraints, how sensitive is the method to errors or noise in the mask?
3.The proposed framework involves Schrodinger bridge sampling with multiple rollout steps and projection operations. Could the authors provide more details about training/inference time and memory requirements compared to diffusion-based editors?
4.Besides standard metrics such as FID, have the authors conducted expert or clinical evaluations to assess whether the generated stenosis patterns are medically realistic?
5.The paper applies GPG periodically during the bridge rollout. How sensitive is performance to the frequency of these guidance steps?

**Limitations:**

Yes

**Strengths And Weaknesses:**

Soundness: The technical part is overall sound. Modeling local edit as optimal transport problem provide a new angle. The proposed GPG solves the soft guidance problem in diffusion-based method. The authors perform extensive experiments on multiple datasets and detectors to verify its effectiveness. However, I think the proposed method relies heavily on geometric mask and vessel prior, which may limit its generalizability.

Presentation: The submission is overall well structured, easy to follow. The literature review is adequate. However, some mathematical inductions are difficult to follow in some part, maybe a few more figures can better explain the mechanism and improve clarity. The GPG part and optimization are not described detailed enough, which may harm reproducibility.

Significance. The paper is addressing a significant problem, lack of diseased labeled data in coronary angiogram. However, since it relies heavily on geometric and vessel prior, it is not clear if it can be generalize to other organs or other domains.

Originality: Optimal transport, Schrodinger bridge and geometric guidance are combined for medical image editing and synthesis. However, I personally think it is more like a combinatory innovation rather than ground-new paradigm shift.

---

> ### Author Rebuttal · Authors · 2026-03-31
>
> We thank you for your detailed assessment of our manuscript. We appreciate your recognition of the effectiveness of our method, as well as your positive comments on the paper organization, literature review, and significance of the study. We also agree that some claims can be further strengthened with additional experiments. We address your points below.
>
> ---
>
> > ## **Q1 & Weaknesses in Soundness & Significance: Generalization w/o Structural Priors**
>
> **A1:** We thank you for this important question.
>
> Our method uses explicit spatial priors, and removing them makes controllable local pixel-level editing harder. More specifically, the framework does not require vessel-specific priors, but it does need a spatial or geometric cue to define where to edit and what shape to follow.
>
> To support this point, we tested our method on LiTS, IDRiD, and ISIC for liver tumor, retinal lesion, and skin lesion editing. Although these tasks not use vessel topology as in CAG, they still provide lesion-region priors. The qualitative results in the https://anonymous.4open.science/r/ICML-3021/generalization.png suggest that OT-Bridge Editor can generalize beyond vessel-specific tasks, as long as some spatial prior is available.
>
> We also agree that, without explicit structural or location priors, controllable local editing becomes harder. This is an important direction for future work.
>
> ---
>
> > ## **Q2 Mask Noise Sensitivity**
>
> **A2:** We agree that the editing mask is a key part of our framework, so its sensitivity to noise should be tested. We added a robustness study with boundary jitter, erosion/dilation, and spatial displacement. The results show that our method remains robust to imperfect masks: although performance drops, synthetic from noisy masks still outperform the real-only baseline.
> |Mask Perturbation|Dice ↑|mAP@50 ↑|
> |-|-:|-:|
> |None|0.774|0.662|
> |Boundary Jitter|0.767|0.645|
> |Erosion/Dilation|0.772|0.587|
> |Spatial Displacement|0.764|0.521|
>
> ---
>
> > ## **Q3 Computational Overhead**
>
> **A3:** We thank you for this important question. We agree that our method adds extra cost due to iterative SB sampling and GPG. We therefore added a table comparing training/inference time, and GPU memory with diffusion-based method.
>
> All methods were tested using 1k denoising steps on L20 GPUs. As expected, our method is more costly, especially with GPG. Still, the cost is practical for offline synthetic data generation, and we believe it is justified by the gains in edit quality, geometric accuracy, and downstream detection performance.
> |Method|GPG|Training(h)|Inference(s/img)|GPU(GB)|
> |-|-|-:|-:|-:|
> |SDM|✗|18.6|104.2|13.4|
> |SiameseDiff|✗|21.3|127.8|14.1|
> |OT-Bridge Editor|✗|24.8|151.6|15.3|
> |OT-Bridge Editor|✓|28.9|179.4|16.7|
>
> ---
>
> > ## **Q4 Clinical Realism Evaluation**
>
> **A4:** We agree that FID alone is not enough. We therefore conducted a reader study: two clinicians classified 100 real and 100 synthetic cases in random order. The synthetic cases were not easy to distinguish from real ones, supporting their clinical realism.
> $$
> Discrimination  Acc (\\%) =
> \\frac{N_{R \\rightarrow R} + N_{S \\rightarrow S}}{N_R + N_S} \\times 100
> $$
> where $R$ denotes real cases and $S$ denotes synthetic cases.
> | Clinician|Discrimination Acc (%) ↓|
> |-|-:|
> |1|58.5|
> |2|61.0|
> |Avg|59.8|
>
> ---
>
> > ## **Q5 GPG Frequency Sensitivity**
>
> **A5:** We thank you for this important question. **Table 6** already shows the benefit of **GPG** with **3 guidance steps**. We further tested **1, 2, 3, 5, and 7 steps** during the SB rollout. The results show that performance is not very sensitive in a moderate range: more guidance helps at first, but the gain becomes small when guidance is too frequent. We therefore keep 3 steps as the default for a good balance between performance and cost.
> |GPG Steps|bDice ↑|bIoU ↑|
> |-|-:|-:|
> |1|0.797|0.725|
> |2|0.870|0.781|
> |3|**0.895**|**0.812**|
> |5|0.887|0.803|
> |7|0.861|0.768|
>
> ---
>
> > ## **Weaknesses in  Presentation**
>
> **A6:** We thank you for this helpful suggestion. We agree that the math part, especially GPG and the optimization steps, can be clearer. We have added a more detailed figure in the Appendix, and we will further revise the main paper with clearer step-by-step explanations and more figures. In short, GPG is a path-level geometric control added during the SB rollout. It guides intermediate states toward the feasible edit region, reduces drift, and improves boundary alignment.
>
> > ## **Weaknesses in  Originality**
>
> **A7:** We appreciate this thoughtful comment. We agree that our work is not a fully new generative paradigm. However, it is more than a simple combination of OT, SB, and geometric guidance. The key novelty is adding set-controlled, path-level guidance inside the stochastic SB process, and using it to cast localized medical image editing as constrained entropic OT with strict geometric constraints. To the best of our knowledge, this is the first such formulation for pixel-accurate localized medical image editing.

---

> > ### Author Rebuttal · Reviewer_nxtU · 2026-04-03
> >
> > I would like to thank the authors for answering my concerns. However, there are a few more details need to confirm:
> >
> > In Q2 Mask Noice Perturbation, how much perturbation (the strength/setting of boundary jitter, erosion etc) is added to the mask? This is important to consider when judging robustness. It would be better to show with images/links.
> >
> > In Q4, Clinical Realism Evaluation, how many doctors are assessed? The detail of the human experiment needs to be shown, e.g., year of experience, expertise etc. It is also better show image examples.
> >
> > In Weaknesses in Presentation, I would like to see the revised pictures and method explanations.

---

> > > ### Author Response · Authors · 2026-04-06
> > >
> > > Thank you very much for your careful and rigorous reading of our rebuttal and for giving us the opportunity to provide more experimental details.
> > >
> > > We fully understand your concern that the rebuttal did not provide sufficient experimental details or the revised figures/method explanations. Due to the strict rebuttal length limit, we had to omit some details. We therefore clarify them below.
> > >
> > > For **Q2**, we agree that the robustness results are only meaningful when the perturbation strength is clearly stated and shown with images. Below, we give the exact settings used in the mask-noise study. We will also add visual examples. Example images are available at https://anonymous.4open.science/r/ICML-3021/perturbations.png .
> > >
> > > We used three types of perturbations:
> > >
> > > 1. **Boundary jitter.**
> > >    We first define a narrow boundary band as the symmetric difference between the dilated mask and the eroded mask, using an elliptical kernel of size `2 × jitter-band + 1`. Inside this band, each pixel is randomly flipped with probability `jitter-prob`.
> > >    Our **main robustness results** use the **gentler setting**: `jitter-band = 1` and `jitter-prob = 0.15`.
> > >    For reference, we also tested a **stronger setting**: `jitter-band = 2` and `jitter-prob = 0.35`.
> > >
> > > 2. **Erosion / dilation.**
> > >    We apply binary erosion or dilation with an odd elliptical kernel. In the quantitative study, the reported morphology result is the **average of erosion and dilation**.
> > >    Our **main robustness results** use the **gentler setting**: `kernel = 3` and `iterations = 1`.
> > >    For reference, we also tested a **stronger setting**: `kernel = 5` and `iterations = 1`.
> > >
> > > 3. **Spatial displacement.**
> > >    We shift the whole editing mask using nearest-neighbor warping with background value 0.
> > >    Our **main robustness results** use the **gentler setting**: `dx = 3` and `dy = 0` pixels.
> > >    For reference, we also tested a **stronger setting**: `dx = 8` and `dy = 0`.
> > >
> > > The quantitative table in the rebuttal uses the **gentler setting**. We think this setting is closer to reasonable annotation or segmentation errors in real use. The stronger setting gives much worse masks, so we use it mainly as an extreme test case, not as a common clinical case. We will make this clearer in the revised paper.
> > >
> > > ---
> > >
> > > For **Q4**, thank you for this helpful follow-up. In the rebuttal, we already noted that the reader study involved **two clinicians**, but we agree that this alone is not sufficient and that the human evaluation setup should be described more clearly. You can find image examples at https://anonymous.4open.science/r/ICML-3021/clinician_examples.pdf .
> > >
> > > In our current study, the reader study involved **two PCI clinicians**, with **10 years** and **15 years** of experience, respectively. They independently reviewed **100 real** and **100 synthetic** CAG cases presented in random order and performed a binary classification task (**real vs. synthetic**). Their discrimination accuracies were **58.5%** and **61.0%**, respectively, with an average of **59.8%**. These results suggest that the synthetic cases were not easy to distinguish from real ones.
> > >
> > > We agree that the rebuttal did not provide enough detail about the participating clinicians or the reader-study protocol. The key details are:
> > > - **number of clinicians:** 2
> > > - **expertise:** PCI clinicians
> > > - **years of experience:** 10 and 15 years
> > > - **evaluation setting:** independent review, random presentation, binary classification of real vs. synthetic
> > > - **case number:** 100 real and 100 synthetic CAG cases
> > >
> > > ---
> > >
> > > For Weaknesses in Presentation. Thank you for this helpful follow-up. We agree that the original presentation of the GPG module and the optimization procedure was not detailed enough, especially for readers who want to understand how the geometric constraint is enforced during the SB rollout.
> > >
> > > To make this part clearer, we provide a revised figure here:
> > > `https://anonymous.4open.science/r/ICML-3021/gpg.pdf`
> > >
> > > This figure gives a step-by-step illustration of how GPG is applied during sampling. In particular, it shows that GPG is not only imposed at the final output, but acts on the **intermediate states** along the SB trajectory. At each selected rollout step, the current state is first updated by the SB transition, and then corrected by a geometry-aware projection. This projection encourages the edited region to match the target stenosis boundary while keeping the non-edited region close to the input image.
> > >
> > > We agree that this explanation should have been made more explicit. The revised figure and the step-by-step description above are intended to clarify the mechanism and make the optimization procedure easier to follow and reproduce.

---

### Official Review · Reviewer_CzAW · 2026-03-12

**Soundness:** 3
**Presentation:** 3
**Significance:** 3
**Originality:** 3
**Overall Recommendation:** 4
**Confidence:** 5

**Summary:**

This paper considers a significant issue: limited annotated coronary angiography (CAG) data for stenosis detection. The authors aim to present a central concept: formulating geometry-constrained stenosis editing as a constrained entropic optimal transport problem, solved via a Schrödinger Bridge with path-level geometric guidance (GPG).

The proposed OT-Bridge Editor enforces hard geometric constraints during generation, enabling pixel-accurate lesion insertion while preserving vessel structure outside the edit mask. Experiments on ARCADE and a multi-center dataset show improved geometric fidelity (Table 1) and consistent downstream detection gains (Table 9–10), outperforming GAN and diffusion baselines.

**Compliance With Llm Reviewing Policy:**

Affirmed.

**Final Justification:**

My overall assessment remains unchanged. The absence of real clinical evaluation significantly reduces my enthusiasm for the paper, as it limits the ability to assess the practical relevance and real-world impact of the proposed method.

**Key Questions For Authors:**

1.	Clinical Relevance: Does synthetic editing preserve physiologically meaningful properties (e.g., % diameter stenosis or FFR consistency)?
This would strengthen clinical impact.
2.	Generality: Can the framework extend to other medical image editing tasks beyond CAG?
This would increase methodological significance.
3.	Robustness: How does performance degrade with noisy or imperfect vessel masks?

**Limitations:**

Yes, to a very minimal extent

**Strengths And Weaknesses:**

Strengths
1.	Clear Motivation – Addresses data scarcity in CAG stenosis detection with a clinically relevant augmentation strategy.
2.	Principled Formulation – Recasting editing as constrained entropic OT with path-level geometric supervision is technically sound and distinct from soft diffusion guidance.
3.	Strong Geometric Control – Superior boundary alignment and structure preservation (Tables 1, 5, 6).
4.	Robust Downstream Gains – Consistent improvements across multiple detectors and datasets (Tables 9–11).
5.	Comprehensive Baselines – Controlled comparisons against GAN and diffusion editors.

Weaknesses
1.	Limited Clinical Validation – Improvements are shown in detection metrics only; no evaluation of stenosis severity estimation, %DS accuracy, or FFR-related impact.
2.	Domain Specificity – Validation is limited to CAG; generalization to other medical editing tasks is unclear.
3.	Incremental Novelty vs Prior SB Work – The distinction from prior Schrödinger Bridge image-to-image methods could be clarified.
4.	Dependence on Accurate Masks – Sensitivity to imperfect segmentation supervision is not evaluated.

---

> ### Author Rebuttal · Authors · 2026-03-31
>
> We are grateful for your insightful review and your positive assessment of the motivation and effectiveness of our method.
>
> We include visualizations at the following link: https://anonymous.4open.science/r/ICML-3021/. Here are our responses to your questions and the identified weaknesses:
>
> ---
>
> > ## **Q1 Clinical Relevance & W1 Limited Clinical Validation**
>
> **A1:**
> We thank you for this important comment. We agree that the current submission places greater emphasis on detection performance, while the evaluation on clinically interpretable metrics is relatively limited. But our experiments already provide evidence for the clinical relevance of the proposed editor. In **Table 1**, we show that our method enables pixel-accurate supervision for coronary stenosis editing, allowing controlled manipulation of stenotic lesions while preserving vessel geometry. In addition, **Appendix A.2, Table 7** reports the severity distribution of the synthesized data, where the stenosis severity bins are defined consistently as: **25–49% DS = mild, 50–69% DS = moderate, 70–99% DS = severe, and 100% DS = occlusion.**
>
> To further strengthen the clinical validation, we additionally generated 1,000 synthetic samples with severity labels following the same %DS-based definition used in the paper. We evaluated coronary stenosis severity estimation, and obtained the following results:
> | Severity Level (%DS) | F1 ↑ | mAP@50 ↑ |
> |-|-:|-:|
> | Mild |0.736|0.684|
> | Moderate |0.787|0.739|
> | Severe |0.821|0.781|
>
> These results further support that our synthetic data are not only useful for lesion detection, but also beneficial for clinically meaningful severity level evaluation.
>
> ---
>
> > ## **Q2 Generality & W2 Domain Specificity**
>
> **A2:** We thank you for the suggestion on broader applicability. Although our current implementation is developed in a vessel-structure composite domain, the structural guidance in our framework can be readily extended by replacing the vessel segmentation map with task-specific segmentation maps from other medical imaging domains. To validate this, we conducted additional experiments on **LiTS[1], IDRiD[2], and ISIC[3]** for **liver tumor, retinal lesion, and skin lesion editing**, respectively. We report **Dice** as a measure of editing accuracy; the **qualitative results are shown in** https://anonymous.4open.science/r/ICML-3021/generalization.png . These results provide initial evidence that our method is not restricted to CAG.
> |Dataset|Dice ↑|
> |-|-:|
> |LiTS|0.802|
> |IDRiD|0.761|
> |ISIC|0.746|
>
> ---
>
> > ## **Q3 Robustness & W4 Dependence on Accurate Masks**
>
> **A3:** We thank you for raising this important concern. In response, we have **added a new robustness study under imperfect segmentation supervision**. Specifically, we perturb the supervision masks with different levels of boundary jitter, erosion/dilation, and spatial displacement, and evaluate both the editing quality and the downstream detector performance.
>
> The results indicate that our method degrades gracefully under mask perturbations. **Dice remains relatively stable under all perturbations**, suggesting that geometric editing quality is not heavily affected by moderate mask noise. However, downstream detection performance drops more noticeably, especially under Spatial Displacement, which is the most damaging perturbation. This suggests that **localization accuracy is more important than minor boundary inaccuracies for our framework**.
>
> |Mask Perturbation|Dice ↑|F1 ↑|mAP@50 ↑|
> |-|-:|-:|-:|
> |None|0.774|0.737|0.662|
> |Boundary Jitter|0.767| 0.717 | 0.645 |
> |Erosion/Dilation|0.772| 0.642 | 0.587 |
> |Spatial Displacement|0.764 | 0.561 | 0.521 |
>
> ---
>
> > ## **W3 Incremental Novelty vs Prior SB Work**
>
> **A4:** We thank you for raising this important point. We agree that prior Schrödinger Bridge (SB) methods have already been explored for image-to-image translation. Accordingly, our contribution is not the generic application of SB to image editing. Rather, our novelty lies in reformulating localized medical image editing as a constrained entropic optimal transport problem with **strict geometric feasibility and path-level supervision**.
>
> Compared with prior SB-based image-to-image approaches, our method differs in **three key aspects**.
> 1. It is designed for localized stenosis editing rather than general i2i, where preserving the unedited anatomical structure is essential.
> 2. It enforces explicit geometric constraints to achieve pixel-level boundary alignment, which is particularly important in medical imaging.
> 3. It introduces GPG supervision (see Appendix Fig. 1), which supervises the generation process along the bridge trajectory, this helps reduce trajectory drift while maintaining high image quality.
>
> ---
>
> [1] Bilic, P., et al. The Liver Tumor Segmentation Benchmark.
>
> [2] Porwal, P., et al. Indian Diabetic Retinopathy Image Dataset.
>
> [3] Codella, N. C. F., et al. Skin Lesion Analysis Toward Melanoma Detection

---

> > ### Author Rebuttal · Reviewer_CzAW · 2026-04-03
> >
> > Thank you for your rebuttal and effort to address my concerns. I still believe that the proposed method warrants evaluation in a clinical setting rather than relying primarily on synthetic experiments. Assessing its performance on real-world clinical data would be essential to establish its practical utility and relevance within the medical domain.

---

> > > ### Author Response · Authors · 2026-04-03
> > >
> > > We sincerely thank you for the positive assessment of our work and for the constructive follow-up comment. We are greatly encouraged by your overall recognition of the paper. At the same time, we believe there may be some **misunderstanding regarding our evaluation setup**, and we would like to clarify this point more explicitly below.
> > >
> > > Our evaluation is not limited to synthetic experiments. Although OT-Bridge generates synthetic vascular images, the paper already includes extensive downstream evaluation **on real clinical data**. Specifically, as described in **Section 4.1 and Appendix A.2.1**, we evaluate on two real datasets: the public ARCADE benchmark and our multi-center internal clinical dataset, which consists of **real intraoperative images collected from patients** across multiple hospital centers. Importantly, the *synthetic images are used only for training augmentation*, while all validation and test evaluations are performed on held-out real images.
> > >
> > > > ### **This means that the reported performance gains reflect improved stenosis detection on real-world clinical data, rather than performance on synthetic samples.**
> > >
> > > Therefore, the evaluation results reported in this paper are **directly relevant** to practical clinical settings.
> > >
> > > > ### **In this sense, our method has indeed been evaluated in a clinically meaningful real-data setting, rather than relying on synthetic-only evaluation. The strong performance on real-world clinical data supports its practical utility in the medical domain.**
> > >
> > > This claim is explicitly supported by the experimental results in the paper. In **Table 3**, on the real ARCADE test set, adding OT-Bridge-generated data consistently improves performance across multiple detectors. For example, with OT-Bridge augmentation, YOLOv8 improves from 0.525 to 0.727 in mAP, DINO-DETR from 0.508 to 0.720, Grounding DINO from 0.276 to 0.418, and RTMDet from 0.545 to 0.675.
> > >
> > > More importantly, the same conclusion holds on our multi-center real-world clinical dataset, which was specifically included to assess generalization in a setting closer to actual medical practice. In Table 4, YOLOv8 improves from 0.654 to 0.731 in mAP, DINO-DETR from 0.638 to 0.725, Grounding DINO from 0.385 to 0.442, and RTMDet from 0.615 to 0.688.
> > > > ### **The paper explicitly states that synthetic data consistently improves detection performance on the multi-center dataset, indicating strong cross-center generalization, which is particularly important in the medical domain.**
> > >
> > > Therefore, we firmly believe that the current work goes beyond purely synthetic evaluation: although the training augmentation data are synthetic, the **evidence of practical utility** is established on public real clinical data and independently collected multi-center real clinical data, as shown in both the main text and the appendix. We fully agree that prospective evaluation in an actual clinical workflow would be an important next step. However, we believe that such a requirement is beyond the intended scope of this paper, which positions OT-Bridge as an offline data augmentation tool, **rather than a deployed clinical decision-making system.**
> > >
> > > If this clarification is consistent with your understanding, **we would be very grateful to know whether it fully addresses your concern.** If so, we would sincerely appreciate it if this could be reflected in your final assessment. Thank you again for your support of our paper.

---

### Decision · Program_Chairs · 2026-04-30

**Decision:**

Accept (regular)

**Comment:**

This paper proposes a framework for generating synthetic coronary angiography (CAG) images with realistic stenosis by formulating localized medical image editing as a constrained entropic optimal transport problem, solved via a Schrödinger Bridge.

Most of the reviewers' concerns were addressed by the rebuttal. For example, limited clinical evaluation, generalization to other domains of medical images, robustness to imperfect masks, incremental novelty by building on existing methods of Schrödinger Bridge and optimal transport, outdated baselines, and computational overhead. These concerns were successfully addressed in the rebuttal.

The remaining concerns include limited scale of clinical study (Reviewer CzAW ), plateau of performance at 1:1 ratio for generative data augmentation (Reviewer PZ3R ), and lack of theoretical guarantee (Reviewer djWH).

To summarize, this paper addresses an important data scarcity problem in coronary angiography with a principled formulation. Consistent gains were achieved with augmented data. Three of four reviewers recommend weak accept. The AC recommends to accept this paper.